# PROVENANCE NETWORKS: END-TO-END EXEMPLAR-BASED EXPLAINABILITY

## ABSTRACT

We introduce *provenance networks*, a novel class of neural models designed to provide end-to-end, training-data-driven explainability. Unlike conventional post-hoc methods, provenance networks learn to link each prediction directly to its supporting training examples as part of the model's normal operation, embedding interpretability into the architecture itself. Conceptually, the model operates similarly to a learned KNN, where each output is justified by concrete exemplars weighted by relevance in the feature space. This approach facilitates systematic investigations of the trade-off between memorization and generalization, enables verification of whether a given input was included in the training set, aids in the detection of mislabeled or anomalous data points, enhances resilience to input perturbations, and supports the identification of similar inputs contributing to the generation of a new data point. By jointly optimizing the primary task and the explainability objective, provenance networks offer insights into model behavior that traditional deep networks cannot provide. While the model introduces additional computational cost and currently scales to moderately sized datasets, it provides a complementary approach to existing explainability techniques. In particular, it addresses critical challenges in modern deep learning, including model opaqueness, hallucination, and the assignment of credit to data contributors, thereby improving transparency, robustness, and trustworthiness in neural models.

## 1 INTRODUCTION

Deep learning has made remarkable progress in recent years, leading to a diverse ecosystem of neural network architectures tailored to specific problem domains (LeCun et al., 2015). Despite this diversity, the vast majority of neural networks share a common design principle: raw input data is transformed through a sequence of nonlinear mappings into an embedding or latent representation. This representation is typically compact, smooth, and task-aligned, making it suitable for downstream tasks. However, in the process of mapping input to such latent spaces, models often lose explicit references to individual training samples. As a result, most networks cannot directly identify which training examples are responsible for shaping a given decision at the test time.

In this paper, we introduce "Provenance Networks", a new class of neural networks (NNs) designed to explicitly trace back predictions to the training data that supports them. At the core of our approach is a neural mechanism that maps any input data point not only to a semantic embedding, but also to an index in the training set. A simplified schematic is shown in Figure 1. From a shared

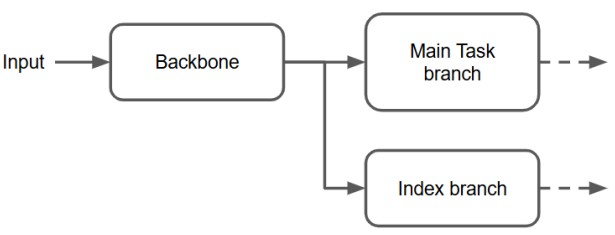

Figure 1: Provenance network schematic.

backbone, one branch handles the primary task, while the other predicts the index of the input sample and is trained jointly during optimization. At inference time, the system retrieves the training sample most likely to have contributed to the prediction. As we will show later, provenance can be trained either as a single dedicated branch or integrated into any existing architecture, enabling

tracking as a standalone task or as an auxiliary component within broader systems for classification, detection, segmentation, or generative modeling.

Our approach combines the interpretability and case-based reasoning of k-Nearest Neighbors (KNNs) with the scalability and representational power of neural networks. This hybrid design enables neighbor-based transparency while supporting efficient end-to-end learning, inference on raw high-dimensional data, and strong generalization. In effect, it allows NNs to implement KNN-like behavior in a fully differentiable, end-to-end manner.

Here, we study the fundamental properties of provenance networks, explore multiple design choices, evaluate their utility across diverse tasks, and analyze their scalability and limitations. Our results demonstrate that provenance networks have broad applicability and significant potential for addressing key challenges in modern AI systems—particularly mitigating hallucinations and enabling fair credit attribution to content creators. Although our experiments focus on the visual domain, the proposed approach is readily applicable to other modalities, including large language models (LLMs).

## 2 RELATED WORK

Since their inception, the black-box nature of neural networks has posed a significant challenge, prompting the development of numerous methods to illuminate their internal workings (Zhang et al., 2021; Lipton, 2018; Linardatos et al., 2020). Existing methods for training data provenance, such as influence functions (Koh & Liang, 2017b) and data Shapley values (Ghorbani & Zou, 2019), provide mathematically rigorous measures of individual sample influence but are computationally expensive and impractical for large-scale datasets. Leave-one-out retraining offers exact influence estimates but is infeasible due to the need for retraining many models (Hammoudeh et al., 2023).

Alternative explainability approaches, including "perturbation-based methods" like LIME (Ribeiro et al., 2016), "game-theoretic methods" such as SHAP (Lundberg & Lee, 2017), and "saliency-based methods" such as vanilla gradients (Simonyan et al., 2014), Integrated Gradients (Sundararajan et al., 2017), SmoothGrad (Smilkov et al., 2017), and GradCAM (Selvaraju et al., 2017), offer feature-level insights into model decision-making. Beyond these, work on "feature visualization and circuits analysis" (Olah et al., 2017; 2020; Zhou et al., 2016) has provided deeper conceptual tools for understanding how neurons, layers, and subnetworks interact, highlighting the compositional structure of representations in neural networks. While these methods highlight which input features or internal mechanisms most influence a prediction, they cannot attribute predictions to specific training samples, limiting their utility for provenance and intellectual property protection.

Current neural information retrieval systems (Mitra & Craswell, 2018; Snell et al., 2017) are used for large-scale classification and retrieval, enabling sample identification. However, these systems face scalability and semantic limitations, and they do not offer a unified framework for controlling memorization and generalization. Provenance Networks address these shortcomings by integrating classification, training data attribution, and robustness within a single model.

Some neural architectures use memory to boost task performance rather than interpretability. Memory networks store and retrieve information for tasks like question answering (Weston et al., 2014; Sukhbaatar et al., 2015), while matching networks enable few-shot learning by classifying new examples based on similarity to a small labeled support set (Vinyals et al., 2016; Xu et al., 2018). Provenance Networks differ fundamentally from memory-augmented models and prototype networks in both purpose and mechanism. Memory-augmented models (*e.g.* Neural Turing Machines Graves et al. (2014)) store and retrieve learned latent memories, which are optimized for task performance rather than interpretability. Prototype networks, similarly, operate on learned class prototypes—compressed centroids that summarize a class rather than referencing specific training points. In contrast, Provenance Networks explicitly retrieve and weight actual training examples, enabling decisions to be grounded in identifiable data instances. This exemplar-level attribution provides transparent, data-driven explanations that neither memory-augmented nor prototype-based architectures can offer.

Interpretability of LLMs has advanced through sparse autoencoders for disentangling latent features (Lieberum et al., 2024) and the broader agenda of mechanistic interpretability (Nanda et al., 2023), while complementary strategies like retrieval-augmented generation (RAG) improve transparency by grounding outputs in external evidence (Lewis et al., 2020).

Building on matching networks and a concept anecdotally noted by Lloyd Watts (link), we expand these ideas through systematic analysis, examining design choices, large-scale dataset strategies, and a range of use cases.

## 3 PROVENANCE NETWORKS

We illustrate the core concepts using multi-class prediction, although they are not limited to this specific task. For a detailed view of the model architectures, please see Appx. 7.1 and 7.2.

### 3.1 I: SINGLE BRANCH NETWORK

Here, essentially each datapoint is mapped to its index (Appx. Fig 10). Inputs may be presented in random order during training, but their indices remain constant. To ease training on large datasets, an input is not always mapped to its own index but is occasionally mapped to a different index from the same class (Appx. Fig. 11). Let $i$ denote the true training index of an input sample $x$, and let $\mathcal{I}_y$ denote the set of indices belonging to the same class $y$, excluding $i$. The target index $t$ for training is sampled according to the mixing parameter $\alpha$ as

$$t \sim \begin{cases} i, & \text{with probability } 1 - \alpha \quad \text{(memorization)} \\ \text{Uniform}(\mathcal{I}_y \setminus \{i\}), & \text{with probability } \alpha \quad \text{(generalization)} \end{cases}. \tag{1}$$

The network outputs logits $\mathbf{p}$ over all training indices, and the cross-entropy loss is then

$$\mathcal{L} = -\log \hat{p}_t,$$

where $\hat{p}_t$ is the predicted probability of the target index $t$ after softmax. This formulation interpolates between pure memorization ($\alpha = 0$) and pure semantic generalization ($\alpha = 1$), with intermediate values controlling the trade-off. This defines a spectrum of model behaviors:

- $\alpha = 0$: pure/rote memorization (*e.g.* index accuracy $\approx 99\%$)
- $\alpha = 1$: pure generalization (*e.g.* semantic accuracy $\approx 100\%$)
- $0 < \alpha < 1$ (*e.g.* $\alpha = 0.3$): balanced behavior (index accuracy $\approx 60\%$, semantic accuracy $\approx 97\%$)

When $\alpha = 1$, the setup effectively reduces to standard classification, with the number of output neurons matching the number of classes. In this case, individual sample identities are lost. Label mixing is applied only during training of the single-branch network, not the two-branch networks.

### 3.2 II: TWO BRANCH NETWORK

We consider two variants of this architecture. In the first variant, called *class-independent*, the main branch predicts the class label $y \in \{1, \ldots, C\}$ and a secondary branch (index branch) predicts an index $z \in \{1, \ldots, K\}$ in dataset. Let $\hat{y}_c$ denote the predicted probability for class $c$ and $\hat{z}_k$ denote the predicted probability for index $k$. Both branches are trained jointly using cross-entropy loss: $\mathcal{L}_{\text{class}} = -\log \hat{y}$ and $\mathcal{L}_{\text{index}} = -\log \hat{z}_z$. The total loss is a weighted sum of the two branches,

$$\mathcal{L}_{\text{total}} = \lambda_{\text{class}} \mathcal{L}_{\text{class}} + \lambda_{\text{index}} \mathcal{L}_{\text{index}} \tag{2}$$

where $\lambda_{\text{class}}$ and $\lambda_{\text{index}}$ control the relative importance of the class and index predictions. We set $\lambda_{\text{class}} = \lambda_{\text{index}} = 1$ in the experiments.

In the second variant, called *class-conditional*, the main branch again predicts the class label $y \in \{1, \ldots, C\}$, but the secondary branch predicts an index *within the predicted class* rather than among all $K$ training samples. Concretely, for a sample belonging to class $y$, the index is defined as

$$z \in \{1, \ldots, K_y\},$$

where $K_y$ is the number of training samples in class $y$. This class-conditional formulation makes index prediction easier, particularly for large datasets where $K \gg K_y$.

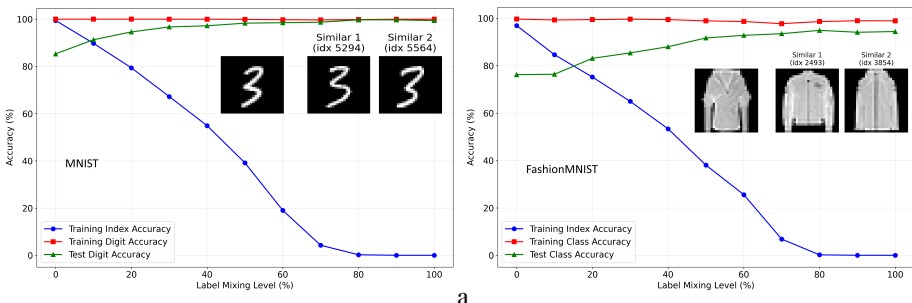

Figure 2: Trade-off between generalization and memorization in the single-branch network, with test samples overlaid alongside their two most similar training examples.

Let $\hat{y}_c$ denote the predicted probability for class $c$, and let $\hat{z}_{k|y}$ denote the predicted probability for index $k$ conditional on class $y$. The corresponding cross-entropy losses are

$$\mathcal{L}_{\text{class}} = -\log \hat{y}_y, \qquad \mathcal{L}_{\text{index}} = -\log \hat{z}_{z|y}.$$

The overall objective is again a weighted combination of the two. At inference time, the model first predicts the class label via the main branch, then uses this class to restrict the index prediction branch to only the indices belonging to that class.

The index branch contains as many neurons as the maximum number of training samples across classes. Alternatively, separate heads can be used per class, in which case the number of neurons in each head matches the number of training samples within that class (*i.e.* no parameter sharing).

If the primary task is not classification, the network must be adjusted to provide a conditioning signal. For instance, in semantic segmentation or image generation, additional outputs can be introduced to predict both the image-level class label (*e.g.* street scene) and the sample index.

## 4 NETWORK ANALYSIS

### 4.1 MEMORIZATION VS. GENERALIZATION TRADE-OFF

Figure 2 shows the results of training the single-branch network on the MNIST (LeCun et al., 1998) and FashionMNIST (Xiao et al., 2017) datasets as the index mixing ratio varies from 0 to 100%. The reported metrics are index prediction accuracy on the training set and class prediction accuracy, derived from the retrieved index, on both training and test sets.

At low levels of label mixing $\alpha$, the network tends to memorize individual samples, which reduces its ability to generalize—evident from the lower test set accuracy. As $\alpha$ increases, memorization decreases and generalization improves. At 100% label mixing, the network completely loses its memorization capacity. This trade-off highlights how one can tune the mixing ratio to balance memorization against generalization for a specific task. A similar trend is observed on FashionMNIST, though in this case memorization has an even stronger negative impact on classification accuracy.

### 4.2 VISUALIZATION OF LEARNED REPRESENTATIONS

We analyze representations learned in the 2048D last embedding layer of the index branch. Figure 3 (top left) shows t-SNE visualizations where misattributed samples (94 out of 10K) lie near cluster boundaries—these samples are visually ambiguous, resembling multiple digit classes. The top-5 retrieved training samples confirm the network organizes data by visual similarity rather than ground truth labels. The bottom row demonstrates instance-level structure: k-means clustering reveals distinct writing styles within digit 6 and dress styles within FashionMNIST, showing the embedding captures fine-grained intra-class variation beyond simple class separation.

### 4.3 SCALABILITY ANALYSIS

To address the fundamental scalability limitation of provenance networks, we investigate whether the system can operate effectively when trained on only a strategically selected subset of training

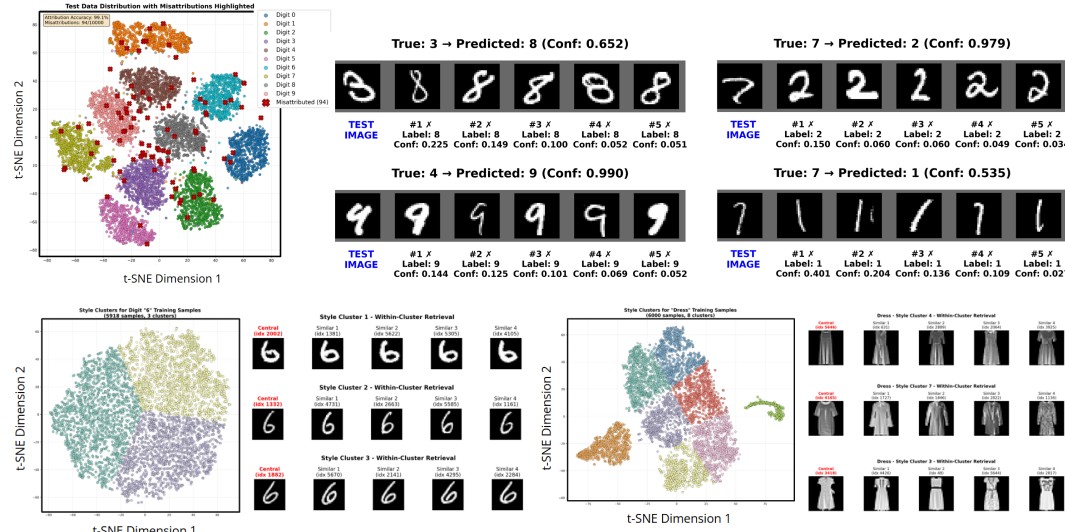

Figure 3: Top (left): t-SNE visualization of the penultimate layer in the index branch of a two-branch class-conditional network. Top (right): Misattributed test samples alongside their five nearest training samples in the index branch. Bottom: t-SNE visualization of k-means clusters from the same layer, with corresponding training samples for digits 6 (left) and FashionMNIST dresses (right).

data. We train both the main and index branches of class-conditional network on identical subsets, selected through stratified sampling to maintain class proportions. Results are shown in Table 1.

Remarkably, training both branches on just 30% of the MNIST data (17,995 samples) achieves 98.87% test accuracy on the main branch—matching the performance of models trained on the full 60K data—while reducing the parameters of the index head by 70%. The index branch attains a Top-5 class matching accuracy of 95.49%, indicating that the network effectively retrieves semantically relevant training examples even when the majority of the data is excluded. On FashionMNIST, training with 50% of the data (30K samples) produces 90.86% accuracy and 89.71% Top-5 accuracy via the index branch, demonstrating consistent performance across datasets. Index prediction accuracy is already high with the full training set (98.16%; Table 2) and reaches 100% using only 50% of the data on MNIST. This suggests that selecting a representative subset can preserve class prediction accuracy while rapidly improving index prediction, enabling the approach to scale.

These results suggest a practical deployment strategy: rather than indexing all training samples, provenance networks can focus on representative exemplars or high-value samples (*e.g.* near decision boundaries, diverse prototypes, or verified clean data). While this compromise means some training samples become unretrievable, it enables substantial parameter reduction while maintaining both classification performance and retrieval capability. This transforms provenance networks from theoretically interesting but impractical to deployable at substantially larger scales. Extended results and detailed analysis are provided in Appx. 7.5.

Table 1: Class prediction accuracy using a two branch class-conditional network on training data subsets.

| Subset | MNIST Samples | Main Branch | Index Branch | | | FashionMNIST Samples | Main Branch | Index Branch | | |
|---|---|---|---|---|---|---|---|---|---|---|
| | | | Top-1 | Top-5 | Top-10 | | | Top-1 | Top-5 | Top-10 |
| 10% | 5,996 | 98.27 | 68.32 | 92.57 | 96.56 | 6,000 | 86.66 | 59.13 | 87.20 | 93.62 |
| 30% | 17,995 | 98.87 | 79.72 | 95.49 | 97.75 | 18,000 | 89.99 | 60.11 | 89.26 | 94.93 |
| 50% | 29,997 | 98.98 | 69.94 | 92.05 | 96.04 | 30,000 | 90.86 | 66.88 | 89.71 | 94.77 |
| 70% | 41,995 | 99.19 | 76.86 | 95.16 | 97.76 | 42,000 | 91.36 | 67.39 | 91.26 | 95.79 |
| 90% | 53,994 | 99.26 | 83.26 | 96.53 | 98.41 | 54,000 | 92.19 | 64.86 | 90.48 | 95.26 |

## 4.4 EFFECT OF MODEL SIZE AND LAYER SHARING

To study how model size impacts generalization vs. memorization in provenance networks, we evaluate two class-conditional models, Small and XLarge, differing in channel dimensions (Small with 4M parameters; XLarge with 80M parameters), and vary the number of shared parameters between

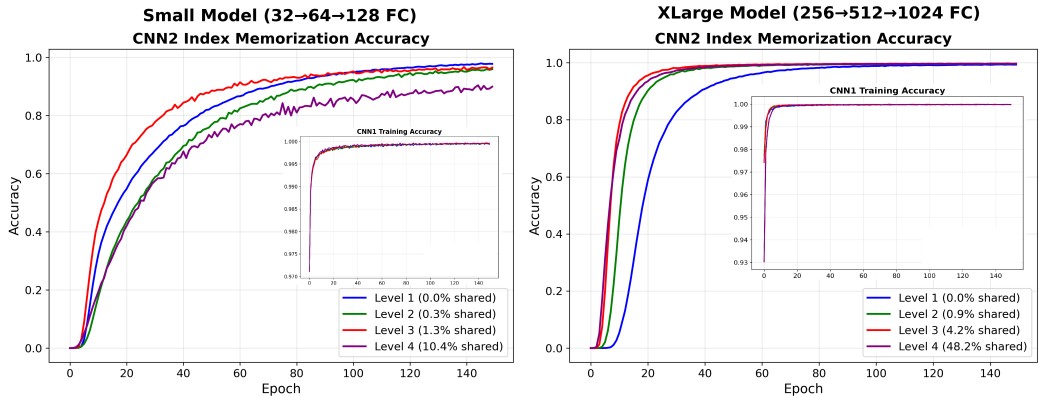

Figure 4: Accuracy per epoch for the index branch and class branch (insets) of Small (left) and XLarge (right) models. Each curve represents a different level of parameter sharing (on MNIST). See also Appx. 7.6.

the branches across 4 levels: Level I (1st conv layer only), Level II (1st two conv layers), Level III (all 3 conv layers), and Level IV (all conv layers plus the first FC layer). Each model was trained for 150 epochs on MNIST. As shown in Figure 4, the larger model converges faster and achieves higher accuracy in both branches, suggesting that greater capacity benefits provenance networks. Increased layer sharing further improves the larger model but can hurt the smaller one—likely due to competition for limited representational capacity between classification and memorization tasks. Larger models have sufficient capacity to accommodate both objectives. See Appx. 7.6.

## 5 APPLICATIONS

### 5.1 IMAGE AND OBJECT CLASSIFICATION

Table 2 summarizes class and index prediction accuracy on four coarse-grained datasets (MNIST, FashionMNIST, CIFAR-10/100 (Krizhevsky & Hinton, 2009)) and one fine-grained dataset (Stanford Dogs by Khosla et al. (2011)). We compare two-branch networks against single-branch networks trained under two levels of label mixing. See Appx. 7.4 for dataset stats.

We did not heavily optimize the networks for accuracy (*e.g.* through data augmentation). Nevertheless, the class-conditional network achieves strong performance in both classification and index prediction, demonstrating that it can both classify and explain. In contrast, the class-independent network performs poorly on index prediction for the CIFAR datasets, primarily due to the large number of neurons required for 60K training samples. Its relatively strong index prediction on MNIST and FashionMNIST can be attributed to the lower complexity of these datasets. Importantly, this shows that the network still provides meaningful explanations in many cases, with higher explainability for CIFAR-10 than CIFAR-100. Similar conclusion applies to dog classification [[cosine similarity]]

Single-branch network results show that models with stronger memorization (100%) explain better but classify worse than those with weaker memorization (50%), as illustrated in Figure 2. In single-branch networks, effective classification requires a compromise, whereas two-branch networks make it possible to achieve both—though at the cost of larger models and greater computational demands.

**Comparison with Other Explainability Methods:** We compared our approach against influence functions, a practical approximation of Shapley-style analysis (Koh & Liang, 2017a). Rather than retraining a classifier for each leave-one-out scenario to quantify a sample's impact, influence functions estimate this effect efficiently through approximations. The label of the nearest or most influential training sample (using Cosine similarity) is assigned to the test point, and accuracy is averaged over the entire test set. However, this method becomes computationally prohibitive and slow on large datasets, as they require Hessian–vector products and often suffer from numerical instability in deep networks (Basu et al., 2020; Feldman, 2020). Consequently, traditional influence estimation is expensive and frequently unreliable. Following prior work (Yeh et al., 2018; Pruthi et al., 2020), we approximate influence using nearest neighbors in the last-layer (or all-layer) representation space. Implementation details and analysis are provided in Appx. 7.12.

Table 2: Classification and index prediction accuracy across 4 settings: two-branch (class-conditional), two-branch (class-independent), and single-branch networks with two levels of memorization. Idx Acc denotes index prediction accuracy on the training set. A memorization level of 100% corresponds to a label mixing parameter of $\alpha = 0$. For the single-branch network, class prediction (Cls Acc) is derived either from the class of the most active neuron (Top-1) or from the majority class among the five most active neurons (Top-5). The network first predicts indices, and the labels associated with those indices are then used for classification. In the class-conditional setting, indices vary only within each class, whereas in the class-independent setting, they span the entire dataset. The last two columns show comparison with influence functions approach.

| | Two-Branch Net Class Conditional | | Two-Branch Net Class Independent | | Single-Branch Net 100% Memorization | | Single-Branch Net 50% Memorization | | Influence Functions | |
| --- | --- | --- | --- | --- | --- | --- | --- | --- | --- | --- |
| | Cls Acc | Idx Acc | Cls Acc | Idx Acc | Cls Acc Top-1/5 | Idx Acc | Cls Acc Top-1/5 | Idx Acc | All Layers | Last Layer |
| MNIST | 99.08 | 98.16 | 99.41 | 99.41 | 84.6/98 | 100 | 98.8/99.7 | 49.6 | 87.25 | 99.36 |
| FMNIST | 96.01 | 98.68 | 92.65 | 98.63 | 76.6/94.7 | 99.8 | 90.3/96.4 | 48 | 76.36 | 92.08 |
| CIFAR-10 | 83.16 | 99.41 | 75.73 | 89.86 | 30.3/68.3 | 99.7 | 65.1/86.4 | 47.3 | 49.46 | 77.97 |
| CIFAR-100 | 37.14 | 99.2 | 38.20 | 40.28 | 8.0/20.5 | 94 | 17.0/37.4 | 32.7 | 26.24 | 37.10 |
| Stanford Dogs | 82.58 | 46.1 | 65.54 | 84.45 | 8.4/17.8 | 99.5 | 9.3/22.6 | 47.9 | | - |

As indicated in the final two columns of Table 2, this baseline achieves strong results but still falls short of the performance obtained by the index head in our two-branch networks.

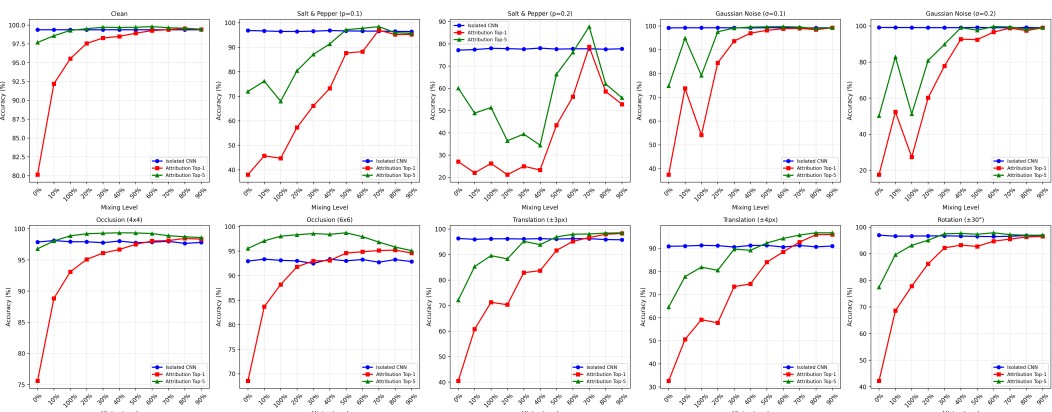

Figure 5: Comparison of the single-branch index-prediction network with varying levels of label mixing against an isolated CNN. Plots show Top-1 and Top-5 accuracy under 9 distortion types plus a baseline without distortion. The variation in the isolated CNN (blue curves) across different index-mixing levels arises from the use of different test sets at each level. Intermediate levels of memorization improve robustness: for distortions like occlusion and blur, partial label mixing (20–30%) yields higher accuracy than the isolated CNN. Performance over remaining 5 distortions is shown in Appx. 7.8.

## 5.2 Robustness to image distortions

We examine whether retrieving similar examples can improve robustness in prediction. To this end, we compare a single-branch index-prediction network with varying levels of label mixing (similar to label smoothing) against a standard CNN trained independently (referred to as isolated CNN in the plots). While the index-prediction network infers indices, the isolated CNN directly predicts class labels. Performance is evaluated under 14 conditions covering 8 distortion types—salt-and-pepper noise, Gaussian noise, occlusion, translation, rotation, scaling, Gaussian blur, and motion blur—along with a 15th baseline condition without distortion. Results on MNIST are presented in Figure 5, with additional results on FashionMNIST provided in Appx. 7.8.

As observed earlier, increasing the level of mixing (hence reducing memorization) generally improves classification accuracy. Interestingly, for certain distortions—such as occlusion, translation, Gaussian, and motion blur—intermediate levels of index mixing yield higher accuracy than the isolated CNN, as indicated by points where the red curve (Top-1 acc) crosses above the blue line (again Top-1 accuracy). This suggests that some degree of memorization (around 20–30%, corresponding to label mixing above 70–80%) can enhance robustness. A possible explanation is that, under cer-

tain distortions (*e.g.* occlusion), the index-prediction network can still retrieve appropriate training samples, whereas a purely classification-based network (isolated CNN) loses this information.

## 5.3 DATASET DEBUGGING

Another application of provenance networks is identifying potentially mislabeled data, outliers, and anomalies by detecting inconsistent or unlikely provenance traces. We apply the two-branch class-conditional network to MNIST and FashionMNIST to detect intra-class anomalies. After training, we compute the entropy of the index-branch output across roughly 6K neurons. For each class, we identify the five training samples with the lowest and highest entropies, shown in Figure 6 for both datasets. Normal samples typically exhibit high entropy, while anomalous or unusual samples show low entropy. This occurs because typical samples activate only a few neurons, whereas atypical samples activate many, making entropy a useful measure for spotting potential outliers. While a standard classification network might also detect anomalies, our approach is complementary, as it leverages instance-level variations captured in the index branch, as illustrated in Figure 3.

Figure 6: Representative (left) and anomalous (right) training samples ranked by index-branch entropy for the two-branch class-conditional network. Low entropy indicates sparse neuron activation. Top rows: MNIST; bottom rows: FashionMNIST. See also Appx. 7.9.

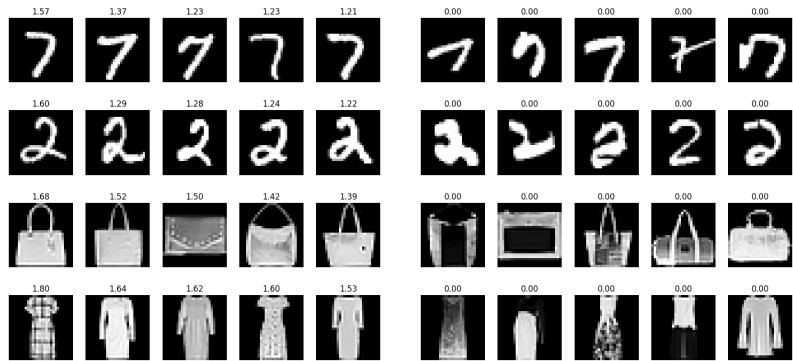

## 5.4 MEMBERSHIP INFERENCE

The objective in this experiment is to determine whether a given input belongs to the training set. We trained the class-conditional two-branch network over four datasets for 40 epochs, during which the index prediction accuracy (top-1 and top-5) reached near-perfect levels across all datasets (*i.e.* overfitted to training indices while maintaining high classification accuracy in the main branch).

To evaluate membership inference, we randomly sampled 5K instances from the training set and 5K from the test set of each dataset. We then computed ROC curves based on the maximum softmax confidence scores from both the class branch and the index branch. As expected, training samples (members) exhibited significantly higher confidence compared to test samples (non-members).

The results across four datasets are presented in Fig. 7. Using the index branch, the AUC was consistently close to perfect. As in previous section, this is because a memorized sample typically activates only one (or a very small subset of) neuron(s), whereas a non-member tends to activate multiple neurons, resulting in lower confidence. In contrast, when using the class branch, the smaller number of output neurons reduces separability based on confidence scores, leading to lower AUC values. Similar trends are observed using entropy (Appx. 7.7).

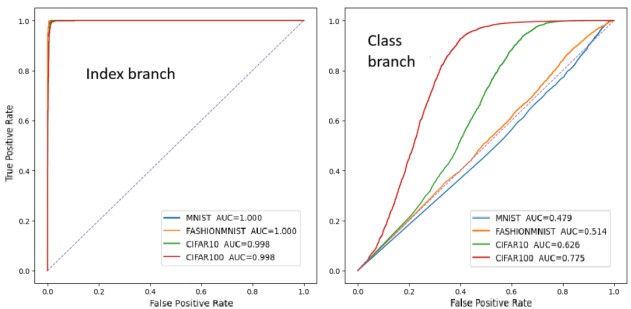

Figure 7: Left: Membership inference results based on the distribution of maximum confidence from the index branch of a class-conditional two-branch network. Right: Corresponding results using the class branch. See also Appx. 7.7.

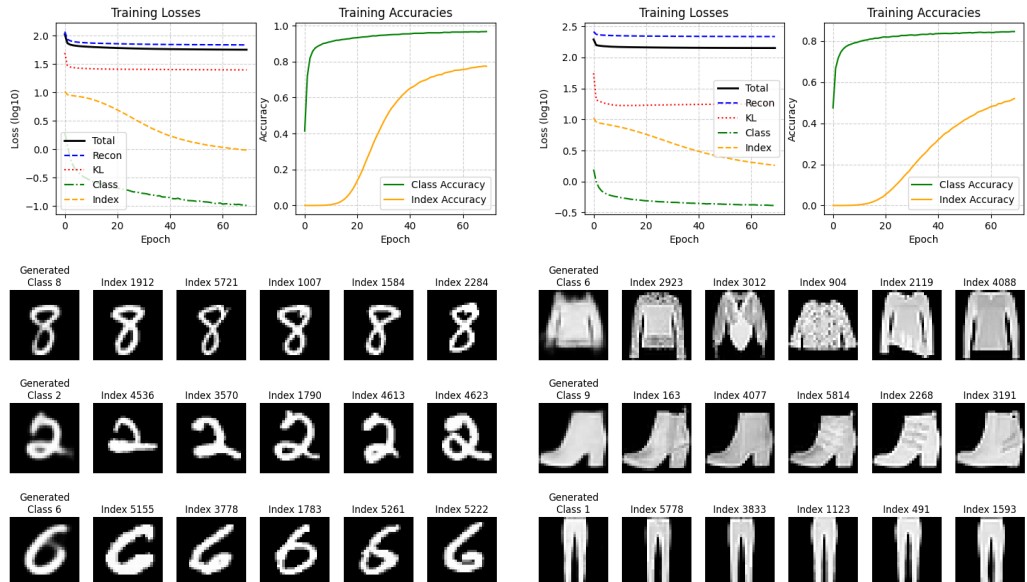

Figure 8: Digit Generation with VAE. The top row displays the training losses and accuracies per epoch, while the bottom row presents generated samples alongside the top-5 predictions from the index prediction network. The left column corresponds to MNIST, and the right column to FashionMNIST (latent_dim=128). We used a simple three-layer U-Net (Ronneberger et al., 2015) as encoder–decoder, with a class head of 10 neurons and 10 index heads each with the numbers of samples in a class (max 6K). See Appx. 7.10 for more details.

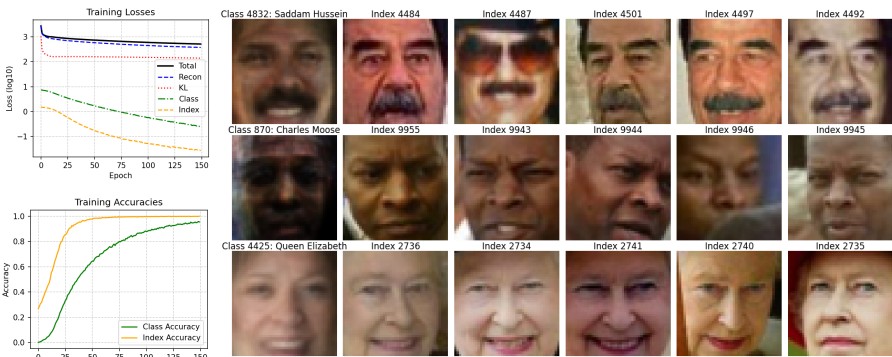

Figure 9: Face generation results over the LFW dataset.

## 5.5 IMAGE GENERATION

The model is a Variational Autoencoder (VAE) (Kingma & Welling, 2014) with two auxiliary supervised heads. The encoder maps input $x_i$ to a latent distribution $(\mu, \sigma)$, from which a latent vector $z$ is sampled. The decoder reconstructs the image from $z$. On an intermediate decoder feature, two classification branches are applied: a class branch predicting $y_i$ and an index branch predicting $k_i$, the sample index within the class. The index branch has one head per class (10 for MNIST). Parameters are not shared across these heads. It is possible to use one index head as in previous experiments. Training minimizes a weighted combination of generative and discriminative objectives:

$$\mathcal{L}_{\text{total}} = \lambda_{\text{gen}}\big(\mathcal{L}_{\text{recon}} + \mathcal{L}_{\text{KL}}\big) + \lambda_{\text{cls}}\big(\mathcal{L}_{\text{class}} + \mathcal{L}_{\text{index}}\big), \tag{3}$$

where $\mathcal{L}_{\text{recon}}$ is the binary cross-entropy reconstruction loss, $\mathcal{L}_{\text{KL}}$ is the KL divergence regularizing the latent space, $\mathcal{L}_{\text{class}}$ and $\mathcal{L}_{\text{index}}$ are cross-entropy losses for class and index predictions, and $\lambda_{\text{gen}}, \lambda_{\text{cls}}$ are weighting factors that balance the generative and discriminative objectives ($\lambda_{\text{gen}} = 0.6, \lambda_{\text{cls}} = 0.4$). This formulation allows the network to simultaneously generate realistic samples while maintaining the ability to classify and retrieve specific training examples. The model is trained with the Adam optimizer (learning rate $10^{-3}$) in mini-batches for 70 epochs, minimizing $\mathcal{L}_{\text{total}}$. Accuracy is tracked for both class and index predictions.

Figure 8 shows loss curves, class and index prediction accuracies for MNIST and FashionMNIST, along with the five closest training samples retrieved by the index branch, demonstrating that the generated samples closely resemble their corresponding training examples.

To evaluate the model on a larger and more complex dataset, we use the LFW face dataset (Huang et al., 2008) containing 13,233 images of 5,749 individuals. We filter to include only persons with up to 25 images, resulting in approximately 10,000-12,000 training images across 4,000-5,000 classes. Each person (class) has a dedicated index prediction head sized to their number of training images. We train the model using a 3-layer fully-connected encoder-decoder architecture with a 100D latent space for 150 epochs ($\lambda_{gen} = 0.6$, $\lambda_{cls} = 0.4$). Accuracy plots in Figure 9 indicate that the model attains high performance on both class and index prediction. Even without extensive hyperparameter tuning, additional loss terms (*e.g.* perceptual loss (Johnson et al., 2016)), or exhaustive optimization, the generated faces exhibit reasonable fidelity, and resemble the retrieved training samples. We found a positive correlation between generation quality and index-prediction confidence, with higher confidence linked to more realistic outputs (more details in Appx. 7.10).

## 6    DISCUSSION AND CONCLUSION

Provenance networks are orthogonal to existing explainability literature. They learn a representation that not only separates classes but also distinguishes individual samples, leading to a better-organized latent space and providing transparency into model decisions.

Provenance networks are relevant to a variety of fields, from intellectual property protection and security to critical applications like healthcare. They enable the tracking of training data, which can help verify copyright, detect attacks like data poisoning, identify outliers, and ensure the reliability of AI systems. In medical imaging, such provenance could assist in identifying dataset biases—such as models relying on spurious hospital-specific artifacts rather than clinical features—though rigorous validation would be required before clinical deployment (*e.g.* by examining similar cases to the input). This transparency is also crucial for regulatory compliance, providing the traceable decisions and data lineage needed to audit AI systems. They also benefit research by providing insight into model behaviors such as hallucination in LLMs and can even be adapted to create faster k-nearest neighbors (KNN) algorithms (Cunningham & Delany, 2021; Zhang et al., 2017).

A key limitation is scalability. As training data grows, index head accuracy drops. This can be mitigated using carefully selected subsets, naturally clustered data, or metadata in unlabeled scenarios, as we showed. The index head also adds computational cost and may impact main-task performance, complicating multi-objective optimization. In the future, we plan to apply our approach to address the hallucination problem in LLMs, to mitigate adversarial vulnerability of neural networks, commercial advertising, and to boost the explainability of other computer vision tasks such as image segmentation and object detection. We will also explore methods to improve the scalability of our approach to larger datasets. Scalability is a common challenge with KNN-like approaches. For instance, influence functions suffer from high computational costs due to approximating or inverting the Hessian matrix, which becomes impractical as datasets and models grow.

A central strength of our method is that it is not limited to explainability; this broader utility distinguishes it from approaches focused only on interpretation. Notably, the index branch also supports applications like dataset reconstruction, with encouraging results that we plan to report in future work.

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

# 7 APPENDIX

## 7.1 SINGLE-STAGE STANDALONE ARCHITECTURE

The standalone (*i.e.* single branch) architecture directly maps features to training sample indices ($\mathbb{R}^m \to \mathbb{R}^N$) without intermediate class structure, representing pure memorization where the model must learn to distinguish between all $N$ training samples simultaneously (Figure 10).

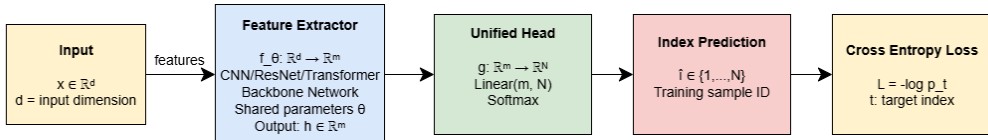

Figure 10: Single-Stage Standalone Architecture for Direct Provenance/Attribution.

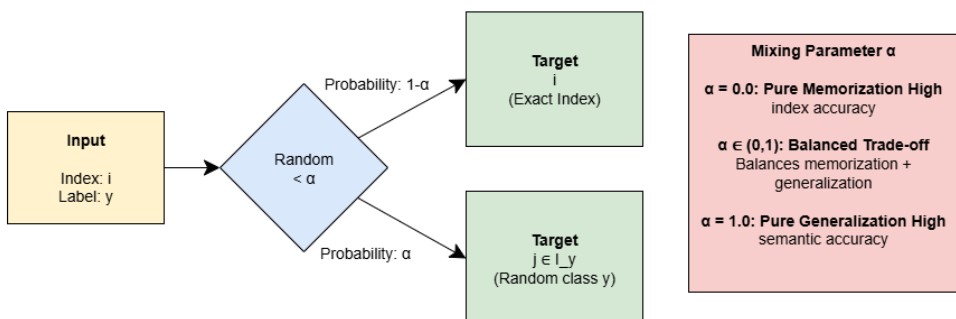

Figure 11: Training Strategy with Mixing Parameter $\alpha$. During training, samples are probabilistically assigned to either exact index targets (memorization) or random class targets (generalization), controlled by mixing parameter $\alpha \in [0,1]$.

The standalone network consists of three primary components: feature extraction, unified attribution head, and direct index prediction, as illustrated in Figure 10.

**Feature Extraction Backbone:** The feature extractor employs a CNN architecture optimized for large-scale memorization tasks:

- **First Convolutional Block:** $\text{Conv2d}(1, 128, 3)$ with padding, BatchNorm2d, ReLU activation, and MaxPool2d(2) reducing spatial dimensions to $14 \times 14$
- **Second Convolutional Block:** $\text{Conv2d}(128, 256, 3)$ with padding, BatchNorm2d, ReLU activation, and MaxPool2d(2) reducing to $7 \times 7$
- **Third Convolutional Block:** $\text{Conv2d}(256, 512, 3)$ with padding, BatchNorm2d, ReLU activation, and AdaptiveAvgPool2d(4, 4) producing fixed $4 \times 4$ spatial output

This configuration yields feature representations $h \in \mathbb{R}^{8192}$ where $8192 = 512 \times 4 \times 4$.

**Unified Attribution Head:** The classification head performs direct mapping from features to training sample probabilities through a deep fully-connected network:

$$h_1 = \text{ReLU}(\text{BN}(\text{Linear}(h, 4096))) \quad \text{with Dropout}(0.4) \tag{4}$$

$$h_2 = \text{ReLU}(\text{BN}(\text{Linear}(h_1, 2048))) \quad \text{with Dropout}(0.2) \tag{5}$$

$$\hat{y} = \text{Softmax}(\text{Linear}(h_2, N)) \quad \text{with Dropout}(0.1) \tag{6}$$

where $N = 60,000$ represents the total number of training samples, and $\hat{y} \in \mathbb{R}^N$ is the probability distribution over all training indices.

**Model Capacity:** The complete architecture contains approximately 129 million trainable parameters, with the final attribution layer contributing $2048 \times 60,000 = 122,880,000$ parameters alone, emphasizing the model's capacity for fine-grained memorization.

### 7.1.1 TRAINING OBJECTIVE AND LOSS FUNCTION

The training objective directly optimizes for exact training sample identification. For each input sample $(x_i, y_i)$ with corresponding training index $t_i$, the model learns the mapping:

$$f_\theta : x_i \mapsto t_i \tag{7}$$

We employ cross-entropy loss with label smoothing ($\epsilon = 0.05$) to stabilize training on the large output space of $N = 60,000$ training samples.

### 7.1.2 OPTIMIZATION STRATEGY

**Optimizer Configuration:** We employ AdamW optimizer with the following hyperparameters:

- Learning rate: $\eta = 0.002$
- Weight decay: $\lambda = 2 \times 10^{-5}$
- Momentum parameters: $\beta_1 = 0.9, \beta_2 = 0.999$
- Batch size: $B = 128$

**Learning Rate Scheduling:** We implement a warmup followed by step decay schedule. The learning rate gradually increases from zero to the base rate over the first 3 epochs. After warmup, we apply step decay every 8 epochs with a multiplicative factor of 0.6, allowing the model to converge effectively.

**Weight Initialization:** Critical for large-scale memorization, we use:

- Convolutional layers: Kaiming normal initialization with mode = fan_out
- Batch normalization: weights = 1, bias = 0
- Final attribution layer: $\mathcal{N}(0, 0.01)$ for enhanced stability
- Other linear layers: $\mathcal{N}(0, 0.02)$

### 7.1.3 EXPERIMENTAL SETUP AND EVALUATION METRICS

We monitor two complementary accuracy metrics during training:

**Index Accuracy:** Measures exact memorization capability **Digit Accuracy:** Measures semantic understanding

**Test Evaluation:** For test samples not present during training, we evaluate both Top-1 and Top-5 digit accuracy.

Training conducted on A100 NVIDIA GPUs with mixed precision (FP16) using PyTorch, enabling efficient memory utilization for the large output space ($N = 60,000$). Models trained for 80 epochs per experiment with comprehensive monitoring of memorization dynamics, convergence patterns, and generalization behavior across different mixing ratios.

**Memorization Experiments:** We conduct comparative analysis between:

- **100% Memorization** ($\alpha = 0.0$): Pure index-level learning
- **50% Memorization** ($\alpha = 0.5$): Balanced memorization-generalization

This experimental design enables systematic investigation of the memorization-generalization trade-off in neural attribution networks and provides insights into the model's capacity for fine-grained training sample identification versus semantic feature learning.

## 7.2 TWO-STAGE PROVENANCE NETWORK

The two-stage provenance network addresses the computational challenges of large-scale attribution by decomposing the problem into hierarchical stages: digit classification followed by instance-level attribution within the predicted class. This approach significantly reduces parameter complexity while enabling conditional attribution based on semantic class structure.

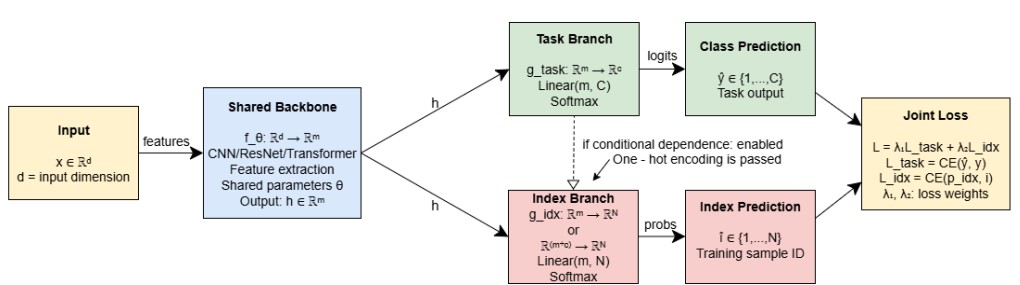

Figure 12: Two-Stage Provenance Network with Optional Class Conditional Dependence. The shared backbone extracts features, which feed into both the task branch (digit classification) and index branch (training sample attribution). When conditional dependence is enabled, the index branch receives concatenated features and one-hot encoded class predictions, allowing instance-level attribution within the predicted class. When disabled, the index branch operates on features alone, performing attribution across all training samples without class-specific guidance

### 7.2.1 ARCHITECTURE OVERVIEW

The two-stage architecture consists of a shared feature extraction backbone feeding into two specialized branches: the task branch for digit classification and the index branch for training sample attribution, as illustrated in Figure 12. The key innovation lies in the conditional dependence mechanism that allows the index branch to leverage class predictions for more focused attribution.

**Shared Feature Backbone:** The feature extraction employs a lightweight CNN architecture:

- **First Block:** $\text{Conv2d}(1, 64, 3)$ with padding, BatchNorm2d, ReLU, $\text{MaxPool2d}(2) \to 14 \times 14 \times 64$

- **Second Block:** $\text{Conv2d}(64, 128, 3)$ with padding, BatchNorm2d, ReLU, $\text{MaxPool2d}(2) \to 7 \times 7 \times 128$

- **Third Block:** $\text{Conv2d}(128, 256, 3)$ with padding, BatchNorm2d, ReLU, $\text{AdaptiveAvgPool2d}(4, 4) \to 4 \times 4 \times 256$

The shared backbone produces feature representations $h \in \mathbb{R}^{4096}$ where $4096 = 256 \times 4 \times 4$, which are then projected to $h' \in \mathbb{R}^{2048}$ through a feature projection layer with BatchNorm and $\text{Dropout}(0.3)$.

**Task Branch (Stage 1):** The digit classification branch performs standard 10-class classification:

$$h_{\text{task}} = \text{ReLU}(\text{BN}(\text{Linear}(h', 512))) \quad \text{with Dropout}(0.2) \tag{8}$$

$$\hat{y}_{\text{digit}} = \text{Softmax}(\text{Linear}(h_{\text{task}}, 10)) \tag{9}$$

where $\hat{y}_{\text{digit}} \in \mathbb{R}^{10}$ represents the digit class probability distribution.

**Index Branch (Stage 2):** The instance attribution branch operates conditionally based on the predicted digit class. The branch architecture depends on whether conditional dependence is enabled:

**Without Conditional Dependence:**

$$h_{\text{idx}} = \text{ReLU}(\text{BN}(\text{Linear}(h', 2048))) \quad \text{with Dropout}(0.2) \tag{10}$$

$$h'_{\text{idx}} = \text{ReLU}(\text{BN}(\text{Linear}(h_{\text{idx}}, 1024))) \quad \text{with Dropout}(0.1) \tag{11}$$

$$\hat{y}_{\text{idx}} = \text{Softmax}(\text{Linear}(h'_{\text{idx}}, M)) \tag{12}$$

where $M$ is the maximum number of samples per class across all digit classes.

**With Conditional Dependence:**

$$h_{\text{concat}} = \text{Concat}(h', \text{OneHot}(\arg\max(\hat{y}_{\text{digit}}))) \tag{13}$$

$$h_{\text{idx}} = \text{ReLU}(\text{BN}(\text{Linear}(h_{\text{concat}}, 2048))) \quad \text{with Dropout}(0.2) \tag{14}$$

$$h'_{\text{idx}} = \text{ReLU}(\text{BN}(\text{Linear}(h_{\text{idx}}, 1024))) \quad \text{with Dropout}(0.1) \tag{15}$$

$$\hat{y}_{\text{idx}} = \text{Softmax}(\text{Linear}(h'_{\text{idx}}, M)) \tag{16}$$

The concatenated input $h_{\text{concat}} \in \mathbb{R}^{2058}$ combines the projected features (2048) with the one-hot encoded predicted digit class (10), enabling class-conditioned attribution.

**Model Capacity:** The two-stage architecture contains approximately 8.7 million parameters, representing a 93.3% reduction compared to the standalone 60K-output model. The parameter distribution includes shared backbone (1.2M), task branch (0.3M), and index branch (7.2M) parameters.

### 7.2.2 TRAINING OBJECTIVE AND MULTI-TASK LOSS

The training objective combines digit classification and instance attribution through a weighted multi-task loss function:

$$\mathcal{L}_{\text{total}} = \alpha \cdot \mathcal{L}_{\text{task}} + \beta \cdot \mathcal{L}_{\text{idx}} \tag{17}$$

where $\alpha = 0.3$ and $\beta = 0.7$ balance the contribution of each task.

**Task Branch Loss:** Standard cross-entropy for digit classification
**Index Branch Loss:** Cross-entropy with label smoothing ($\epsilon = 0.05$) for instance attribution:

### 7.2.3 CLASS-CONDITIONED ATTRIBUTION MECHANISM

**Index Mapping Strategy:** The two-stage approach requires bidirectional mapping between global training indices and class-local indices:

$$\text{global\_to\_local} : \{0, 1, ..., N-1\} \rightarrow \{0, 1, ..., 9\} \times \{0, 1, ..., M_c - 1\} \tag{18}$$

$$\text{local\_to\_global} : \{0, 1, ..., 9\} \times \{0, 1, ..., M_c - 1\} \rightarrow \{0, 1, ..., N-1\} \tag{19}$$

where $M_c$ is the number of samples in digit class $c$, and $M = \max_c M_c$.

**Validity Masking:** During inference, the index branch output is masked to prevent invalid predictions:

$$\hat{y}_{\text{idx}}^{\text{masked}}[j] = \begin{cases} \hat{y}_{\text{idx}}[j] & \text{if } j < M_{\hat{c}} \\ -\infty & \text{otherwise} \end{cases} \tag{20}$$

where $\hat{c} = \arg\max(\hat{y}_{\text{digit}})$ is the predicted digit class and $M_{\hat{c}}$ is the number of training samples in that class.

**Teacher Forcing:** During training, we employ teacher forcing where the index branch uses ground truth digit labels rather than predictions:

$$h_{\text{concat}}^{\text{train}} = \text{Concat}(h', \text{OneHot}(y_i^{\text{digit}})) \tag{21}$$

This stabilizes training by providing accurate class information to the attribution branch.

### 7.2.4 Conditional vs. Non-Conditional Modes

The architecture supports two operational modes:

**Non-Conditional Mode:** The index branch operates independently of class predictions, performing attribution across all training samples without class-specific guidance. Input dimensionality to the index branch remains 2048.

**Conditional Mode:** The index branch receives concatenated features and class information, enabling class-conditioned attribution. Input dimensionality increases to 2058, allowing the model to focus attribution within the predicted semantic class.

The conditional dependence mechanism provides several advantages:

- **Focused Attribution:** Restricts search space to semantically relevant training samples
- **Improved Accuracy:** Leverages class structure for more precise instance matching
- **Computational Efficiency:** Reduces effective output space from $N$ to $\max_c M_c$
- **Interpretability:** Attribution results are constrained to the predicted semantic class

### 7.2.5 Optimization Strategy

**Optimizer Configuration:** AdamW with identical hyperparameters to the standalone model described in Section 7.1.2.

### 7.2.6 Evaluation Metrics

The two-stage architecture requires specialized evaluation metrics for each stage:

**Stage 1 (Digit Accuracy):** Standard classification accuracy **Stage 2 (Instance Accuracy):** Local index prediction accuracy within the ground truth class **End-to-End Attribution Accuracy:** Overall system performance combining both stages

### 7.2.7 Experimental Configuration

**Architecture Variants:** We evaluate both conditional and non-conditional modes to assess the impact of class-guided attribution on overall system performance.

**Computational Efficiency:** The two-stage approach enables efficient batch processing with masked outputs, avoiding the computational overhead of the full 60K-dimensional softmax in the standalone architecture.

**Training Paradigm:** Joint end-to-end training of both branches with shared backbone parameters, enabling the model to learn complementary representations for classification and attribution tasks simultaneously.

This hierarchical decomposition enables scalable attribution learning while maintaining semantic coherence through class-conditioned instance matching, providing a computationally efficient alternative to direct large-scale memorization approaches.

### 7.3 ANALYSIS OF LEARNED EMBEDDINGS

Figure 13 displays a t-SNE visualization of k-means clusters generated from the penultimate layer of the index branch of a two-branch, class-conditional network. The architecture is detailed in Section 7.2. The analysis was conducted on 5,842 training samples of the digit '4'. For each cluster center, the four closest training data points, selected based on Euclidean distance in the feature space, are shown to illustrate the cluster's composition.

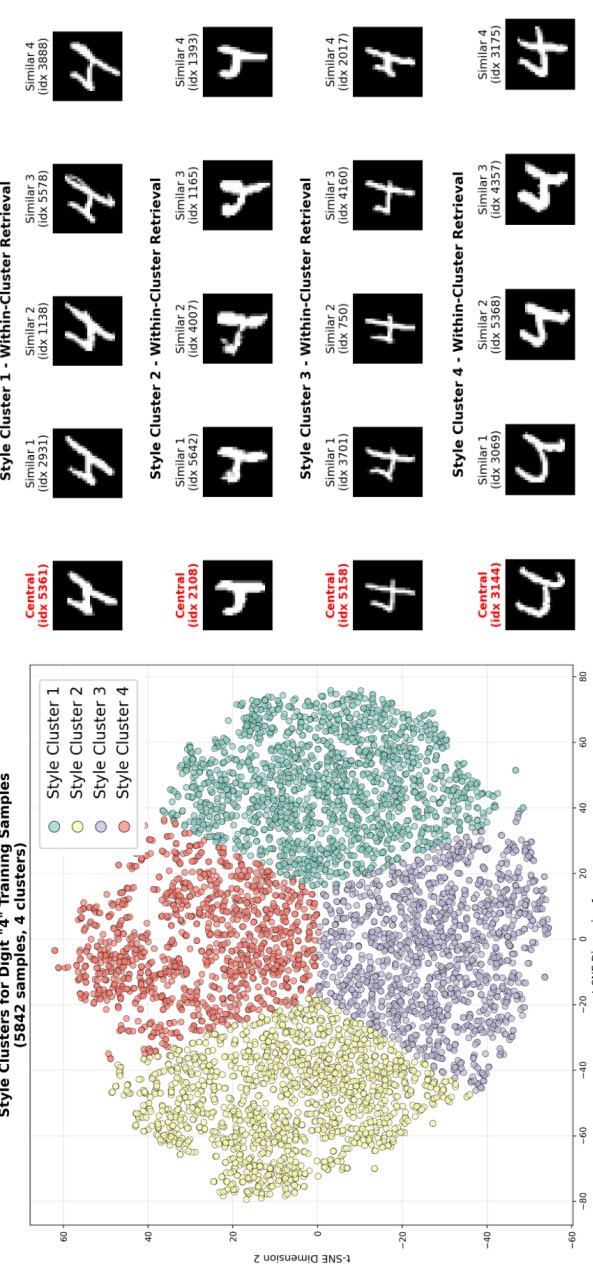

Figure 13: Different styles for digit 4 derived from K-means clustering of penultimate layer of the index branch of a two-branch class-conditional network.

## 7.4 DATASETS

We evaluate provenance networks across a diverse collection of datasets spanning different visual domains, complexity levels, and dataset sizes. Our experimental design progresses from simple grayscale digit recognition to complex natural image classification, enabling systematic analysis of how provenance networks scale across different visual domains and dataset complexities.

### 7.4.1 COMPUTER VISION DATASETS

**MNIST** (LeCun et al., 1998): The Modified National Institute of Standards and Technology database contains 70,000 grayscale images of handwritten digits (0-9) at 28×28 pixel resolution. We use the standard split of 60,000 training samples and 10,000 test samples. Each digit class contains approximately 6,000 training examples, with slight variations across classes. The dataset serves as our primary testbed for fundamental provenance network analysis due to its manageable size and clear class structure.

**Fashion-MNIST** (Xiao et al., 2017): A direct replacement for MNIST consisting of 70,000 grayscale images of fashion items across 10 categories (T-shirts, trousers, pullovers, dresses, coats, sandals, shirts, sneakers, bags, ankle boots). The dataset maintains the same 28×28 resolution and 60,000/10,000 train/test split as MNIST but presents significantly higher visual complexity with greater intra-class variation and inter-class similarity, making it more challenging for both classification and attribution tasks.

**CIFAR-10** (Krizhevsky & Hinton, 2009): A collection of 60,000 32×32 color images across 10 object classes (airplane, automobile, bird, cat, deer, dog, frog, horse, ship, truck). The standard split provides 50,000 training images and 10,000 test images, with 5,000 training samples per class. CIFAR-10 represents a significant complexity increase from the grayscale datasets, featuring natural images with complex backgrounds, lighting variations, and object poses.

**CIFAR-100** (Krizhevsky & Hinton, 2009): An extension of CIFAR-10 containing 60,000 32×32 color images across 100 fine-grained classes grouped into 20 coarse categories. With only 500 training samples per class, CIFAR-100 presents substantial challenges for memorization-based approaches while testing the scalability of provenance networks to larger class vocabularies and reduced per-class sample sizes.

**Stanford Dogs** (Khosla et al., 2011): A fine-grained classification dataset containing approximately 20,580 images across 120 dog breeds. Images vary significantly in resolution and aspect ratio, presenting challenges in both visual complexity and fine-grained discrimination. The dataset tests provenance networks' ability to handle real-world image variation and subtle inter-class differences that require detailed visual understanding.

**Labeled Faces in the Wild (LFW)** (Huang et al., 2008): A face recognition dataset containing over 13,000 images of faces collected from the web, with significant variation in pose, lighting, expression, and image quality. We use LFW to evaluate provenance networks in generative modeling tasks, specifically testing whether generated faces can be traced back to their most similar training examples.

### 7.4.2 DATASET STATISTICS AND CHARACTERISTICS

Table 3 summarizes the key characteristics of each dataset used in our experiments. The progression from MNIST to Stanford Dogs represents increasing visual complexity, class granularity, and real-world applicability.

**Complexity Considerations:** The datasets are strategically selected to evaluate different aspects of provenance networks:

- **Scale Testing:** MNIST and Fashion-MNIST provide controlled environments for fundamental algorithm development with manageable computational requirements.

- **Class Granularity:** The progression from 10 classes (MNIST, Fashion-MNIST, CIFAR-10) to 100+ classes (CIFAR-100, Stanford Dogs) tests scalability of both standalone and two-stage architectures.

Table 3: Dataset statistics and characteristics for provenance network evaluation

| Dataset | Classes | Train Size | Test Size | Resolution | Channels | Complexity |
|---|---|---|---|---|---|---|
| MNIST | 10 | 60,000 | 10,000 | 28×28 | 1 | Simple |
| Fashion-MNIST | 10 | 60,000 | 10,000 | 28×28 | 1 | Moderate |
| CIFAR-10 | 10 | 50,000 | 10,000 | 32×32 | 3 | Moderate |
| CIFAR-100 | 100 | 50,000 | 10,000 | 32×32 | 3 | High |
| Stanford Dogs | 120 | 12,000 | 8,580 | Variable | 3 | High |
| LFW | 5,749 | 13,000 | Variable | Variable | 3 | High |

- **Visual Complexity:** Moving from grayscale digits to natural color images evaluates the robustness of learned representations across visual domains.
- **Sample Density:** CIFAR-100's 500 samples per class versus MNIST's 6,000 samples per class tests performance under varying data availability.
- **Fine-grained Recognition:** Stanford Dogs' subtle inter-class differences challenge the attribution system's ability to capture discriminative features.

### 7.4.3 DATA PREPROCESSING AND NORMALIZATION

All datasets undergo consistent preprocessing to ensure fair comparison across architectures:

**Normalization:** Images are normalized using dataset-specific statistics:

- MNIST/Fashion-MNIST: $\mu = 0.1307, \sigma = 0.3081$
- CIFAR-10/100: $\mu = (0.4914, 0.4822, 0.4465), \sigma = (0.2023, 0.1994, 0.2010)$
- Stanford Dogs/LFW: ImageNet statistics for transfer learning compatibility

**Data Augmentation:** We deliberately avoid extensive data augmentation in our primary experiments to maintain direct correspondence between augmented samples and their training indices. This design choice preserves the integrity of the attribution task, where each training sample must maintain a unique, identifiable index.

**Resolution Handling:** For datasets with variable resolutions (Stanford Dogs, LFW), images are resized to consistent dimensions while maintaining aspect ratios through center cropping or padding as appropriate.

### 7.4.4 EXPERIMENTAL PARTITIONS

**Training Set Attribution:** During training, each sample in the training set is assigned a unique index $i \in \{0, 1, ..., N - 1\}$ where $N$ is the total number of training samples. These indices remain constant throughout training, enabling the provenance network to learn stable index-to-sample mappings.

**Validation and Testing:** Test sets are used exclusively for evaluation, with provenance networks tasked to identify the most similar training samples for each test input. This setup simulates real-world scenarios where models must trace novel inputs back to their training data influences.

**Cross-Dataset Generalization:** While our primary focus is within-dataset attribution, the diverse dataset collection enables analysis of how provenance learning principles transfer across visual domains with different statistical properties and semantic structures.

This comprehensive dataset collection enables systematic evaluation of provenance networks across the spectrum from simple digit recognition to complex real-world visual understanding, providing robust evidence for the approach's broad applicability and scalability characteristics.

## 7.5 Scalability Through Subset Sampling: Extended Results

To comprehensively evaluate the scalability approach presented in Section 4.3, we conducted experiments across fine-grained subset ratios from 10% to 100% of the training data. Table 4 presents complete results for MNIST and FashionMNIST.

**Experimental Setup:** We use stratified sampling to maintain class proportions when selecting subsets. Both CNN1 (classification branch) and CNN2 (index branch) are trained on the same subset of training data, with CNN2 predicting indices only within the selected subset. Both models share the initial convolutional layer and are trained jointly for 100 epochs. All models are evaluated on the complete 10,000-sample test set.

**Key Observations:**

**General trends with data scale:** As expected, increasing the subset size generally improves performance, with CNN1 test accuracy improving from 98.27% (10% subset) to 99.26% (90% subset) on MNIST. However, the improvements plateau beyond 50-70%, demonstrating diminishing returns. Notably, even with severely limited subsets (10% = 6,000 samples), CNN1 achieves respectable accuracy (86.66% on FashionMNIST, 98.27% on MNIST), validating that provenance networks can operate effectively when trained on substantially reduced data.

**Non-monotonic index prediction:** CNN2 Top-1 accuracy does not increase monotonically with subset size. For MNIST, Top-1 accuracy peaks at 30% (79.72%) and 90% (83.26%), while dropping at intermediate points (e.g., $50\% \rightarrow 69.94\%$). This counterintuitive pattern suggests that adding more training samples to the index vocabulary introduces confusion between visually similar examples, making exact index prediction harder even as the model has more data. The effect is less pronounced in Top-5 and Top-10 metrics, which remain more stable.

**Stable semantic retrieval:** Top-5 and Top-10 accuracies show much more consistent trends across subset sizes, indicating the network successfully identifies semantically relevant training samples regardless of exact index prediction difficulty. For instance, MNIST Top-5 accuracy ranges from 92.05% to 96.53% across all subsets, with no dramatic drops. This validates the Top-K retrieval strategy for provenance tracking—the network learns to map test samples to their nearest neighbors in the training set, even when pinpointing the exact closest sample proves difficult.

**Dataset complexity effects:** FashionMNIST shows performance saturation beyond 50%, with minimal improvement from additional subset samples (90%: 92.19% vs 50%: 90.86%). This suggests that for more complex datasets with higher intra-class variation, carefully selected representative samples (prototypes, boundary cases, or diversity-maximizing selections) may be more effective than random stratified sampling. The marginal gains from 50% to 90% (1.33 percentage points) come at the cost of nearly doubling the index head size.

**Practical deployment:** For MNIST, training on 30% of data achieves 98.87% classification accuracy with 95.49% Top-5 retrieval, representing 70% parameter reduction in the index head (17,995 vs 60,000 output neurons). For FashionMNIST, training on 50% achieves 90.86% classification with 89.71% Top-5 retrieval and 50% parameter reduction. These results demonstrate practical scalability improvements while maintaining competitive performance. The trade-off between parameter efficiency and accuracy allows practitioners to select operating points based on deployment constraints: resource-constrained settings can use 30-50% subsets with minimal accuracy loss, while applications requiring maximum accuracy can use 70-90% subsets while still achieving meaningful compression compared to full indexing.

Table 4: Scalability analysis: Both classification and index branches trained on the same subset. CNN1 provides test accuracy; CNN2 Top-K shows class matching accuracy of retrieved training samples.

| Subset | Samples | CNN1 | Top-1 | Top-5 | Top-10 |
|--------|---------|------|-------|-------|--------|
| **MNIST** | | | | | |
| 10% | 5,996 | 98.27 | 68.32 | 92.57 | 96.56 |
| 30% | 17,995 | 98.87 | 79.72 | 95.49 | 97.75 |
| 50% | 29,997 | 98.98 | 69.94 | 92.05 | 96.04 |
| 70% | 41,995 | 99.19 | 76.86 | 95.16 | 97.76 |
| 90% | 53,994 | 99.26 | 83.26 | 96.53 | 98.41 |
| **FashionMNIST** | | | | | |
| 10% | 6,000 | 86.66 | 59.13 | 87.20 | 93.62 |
| 30% | 18,000 | 89.99 | 60.11 | 89.26 | 94.93 |
| 50% | 30,000 | 90.86 | 66.88 | 89.71 | 94.77 |
| 70% | 42,000 | 91.36 | 67.39 | 91.26 | 95.79 |
| 90% | 54,000 | 92.19 | 64.86 | 90.48 | 95.26 |

## 7.6 Analysis of network size and parameter sharing

We conducted systematic parameter sharing experiments across multiple model scales and datasets to understand how different levels of parameter sharing affect task performance. All models follow a convolutional neural network architecture with progressive channel expansion, consisting of three convolutional layers followed by fully connected layers.

Table 5: MNIST results across all model sizes and sharing levels. C2 Cls-T1/T5 denotes CNN2 Class Consistency Top-1/Top-5 accuracy.

| Model | Level | Sharing% | Total Params | CNN1 Test | CNN2 Memo (training) | C2 Cls-T1/T5 |
|---|---|---|---|---|---|---|
| Small (32→64→128) | 1 | 0.0% | 4.0M | 0.992 | 0.987 | 0.817 / 0.972 |
| | 2 | 0.3% | 4.0M | 0.992 | 0.982 | 0.820 / 0.972 |
| | 3 | 1.3% | 4.0M | 0.991 | 0.987 | 0.828 / 0.975 |
| | 4 | 10.4% | 3.9M | 0.991 | 0.919 | 0.766 / 0.944 |
| Medium (64→128→256) | 1 | 0.0% | 17.2M | 0.993 | 1.000 | 0.850 / 0.977 |
| | 2 | 0.4% | 17.2M | 0.993 | 1.000 | 0.847 / 0.977 |
| | 3 | 2.1% | 17.1M | 0.993 | 1.000 | 0.855 / 0.978 |
| | 4 | 18.8% | 16.1M | 0.993 | 0.963 | 0.827 / 0.965 |
| Large (128→256→512) | 1 | 0.0% | 35.8M | 0.993 | 0.997 | 0.863 / 0.981 |
| | 2 | 0.7% | 35.7M | 0.994 | 0.997 | 0.861 / 0.981 |
| | 3 | 3.1% | 35.5M | 0.993 | 0.997 | 0.862 / 0.981 |
| | 4 | 31.8% | 33.0M | 0.992 | 0.994 | 0.848 / 0.976 |
| XLarge (256→512→1024) | 1 | 0.0% | 79.6M | 0.993 | 0.992 | 0.869 / 0.982 |
| | 2 | 0.9% | 79.5M | 0.993 | 0.992 | 0.866 / 0.981 |
| | 3 | 4.2% | 79.0M | 0.993 | 0.993 | 0.866 / 0.981 |
| | 4 | 48.2% | 71.9M | 0.992 | 0.994 | 0.852 / 0.978 |

Table 6: Fashion-MNIST results across all model sizes and sharing levels. C2 Cls-T1/T5 denotes CNN2 Class Consistency Top-1/Top-5 accuracy.

| Model | Level | Sharing% | Total Params | CNN1 Test | CNN2 Memo (training) | C2 Cls-T1/T5 |
|---|---|---|---|---|---|---|
| Small (32→64→128) | 1 | 0.0% | 4.0M | 0.895 | 0.933 | 0.555 / 0.820 |
| | 2 | 0.3% | 4.0M | 0.893 | 0.944 | 0.568 / 0.832 |
| | 3 | 1.3% | 4.0M | 0.895 | 0.998 | 0.624 / 0.877 |
| | 4 | 10.4% | 3.9M | 0.887 | 0.826 | 0.521 / 0.787 |
| Medium (64→128→256) | 1 | 0.0% | 17.2M | 0.908 | 0.757 | 0.439 / 0.717 |
| | 2 | 0.4% | 17.2M | 0.906 | 0.977 | 0.595 / 0.850 |
| | 3 | 2.1% | 17.1M | 0.906 | 0.982 | 0.611 / 0.860 |
| | 4 | 18.8% | 16.1M | 0.893 | 0.950 | 0.565 / 0.827 |
| Large (128→256→512) | 1 | 0.0% | 35.8M | 0.909 | 0.987 | 0.618 / 0.868 |
| | 2 | 0.7% | 35.7M | 0.912 | 0.996 | 0.629 / 0.876 |
| | 3 | 3.1% | 35.5M | 0.910 | 0.998 | 0.633 / 0.879 |
| | 4 | 31.8% | 33.0M | 0.905 | 0.998 | 0.622 / 0.872 |
| XLarge (256→512→1024) | 1 | 0.0% | 79.6M | 0.914 | 0.835 | 0.504 / 0.768 |
| | 2 | 0.9% | 79.5M | 0.913 | 0.998 | 0.631 / 0.879 |
| | 3 | 4.2% | 79.0M | 0.914 | 0.998 | 0.635 / 0.882 |
| | 4 | 48.2% | 71.9M | 0.908 | 0.998 | 0.623 / 0.873 |

We evaluated four model sizes with increasing capacity. The Small Model uses a 32→64→128 channel progression with 128 FC units, totaling approximately 4M parameters. The Medium Model expands to 64→128→256 channels with 256 FC units, reaching approximately 17M parameters. The Large Model further scales to 128→256→512 channels with 512 FC units, comprising approximately 35M parameters. Finally, the XLarge Model uses 256→512→1024 channels with 1024 FC units, totaling approximately 80M parameters. All models include dropout (0.5) and ReLU activations.

We implemented four levels of parameter sharing between two networks. Level 1 shares only the first convolutional layer. Level 2 shares the first two convolutional layers. Level 3 shares all three convolutional layers. Level 4 shares all convolutional layers plus the first fully connected layer. This progressive sharing design allows us to study how increasing amounts of shared representations affect task interference and performance.

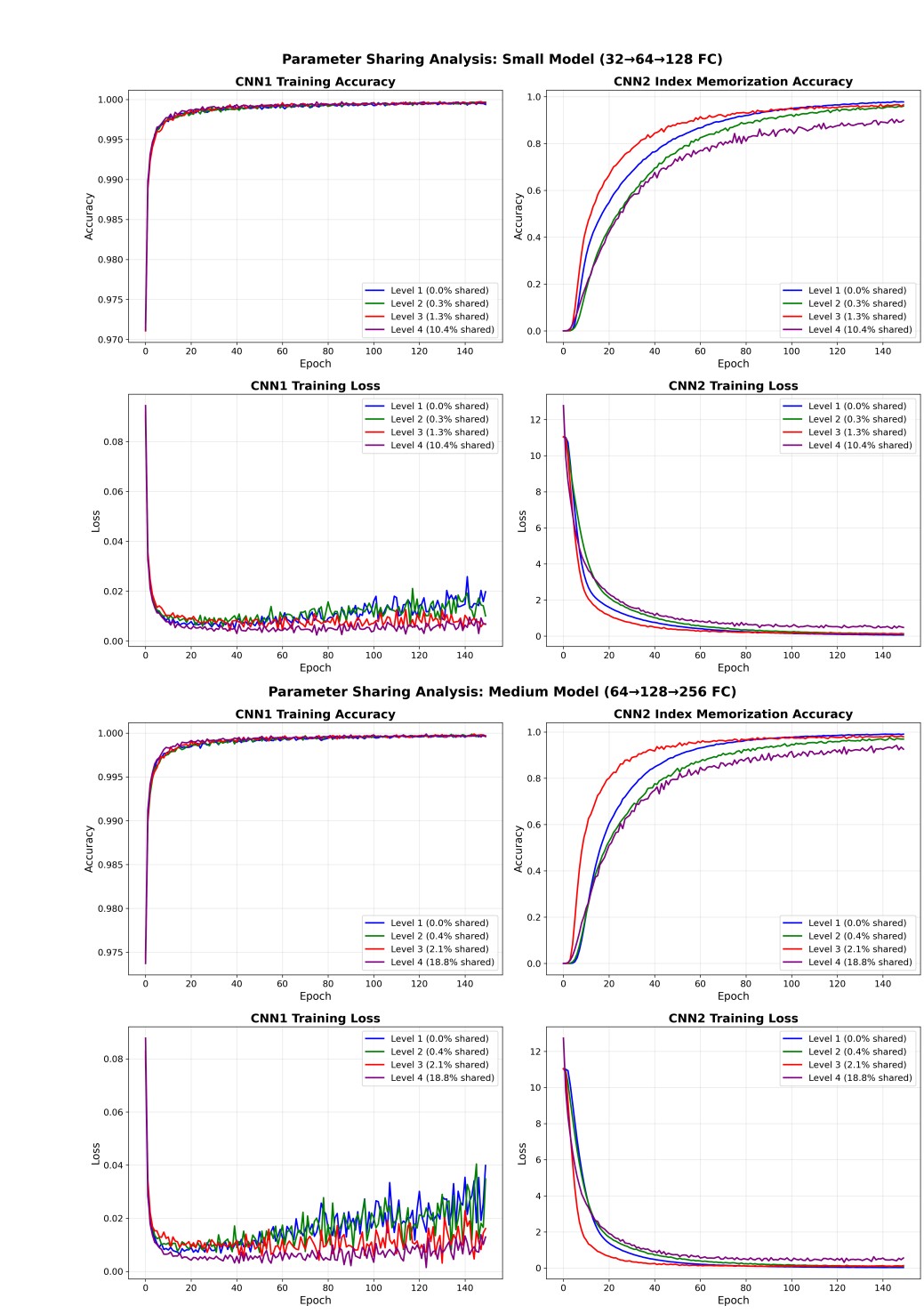

Figure 14: Convergence plots for class branch (CNN1) and index branch (CNN2) for a small (top) and medium (bottom) size CNNs over MNIST dataset.

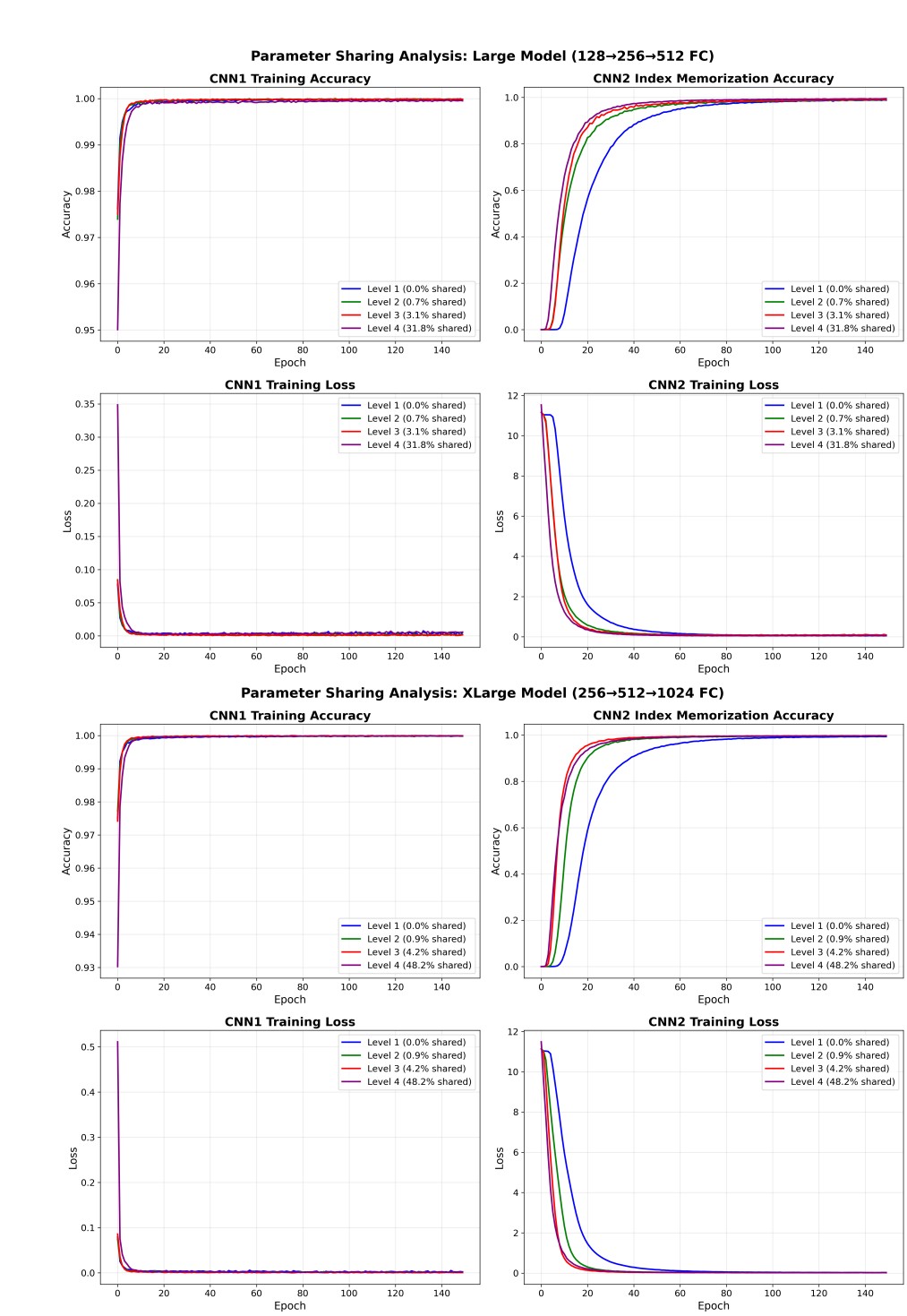

Figure 15: Convergence plots for class branch (CNN1) and index branch (CNN2) for a large (top) and xlarge (bottom) size CNNs over MNIST dataset.

1404
1405
1406
1407
1408
1409
1410
1411
1412
1413
1414
1415
1416
1417
1418
1419
1420
1421
1422
1423
1424
1425
1426
1427
1428
1429
1430
1431
1432
1433
1434
1435
1436
1437
1438
1439
1440
1441
1442
1443
1444
1445
1446
1447
1448
1449
1450
1451
1452

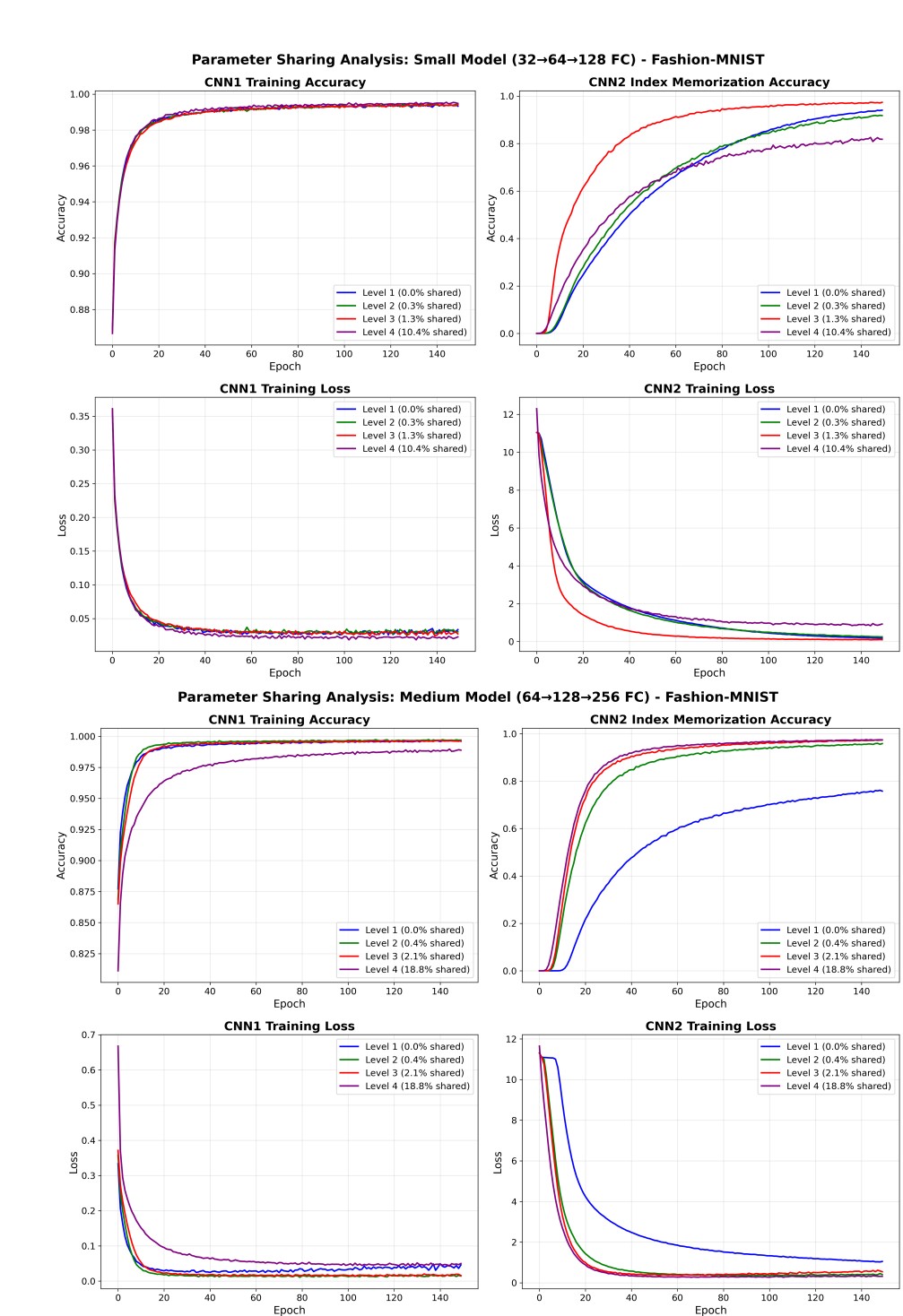

Figure 16: Convergence plots for class branch (CNN1) and index branch (CNN2) for a small (top) and medium (bottom) size CNNs over FashionMNIST dataset.

1453
1454
1455
1456
1457

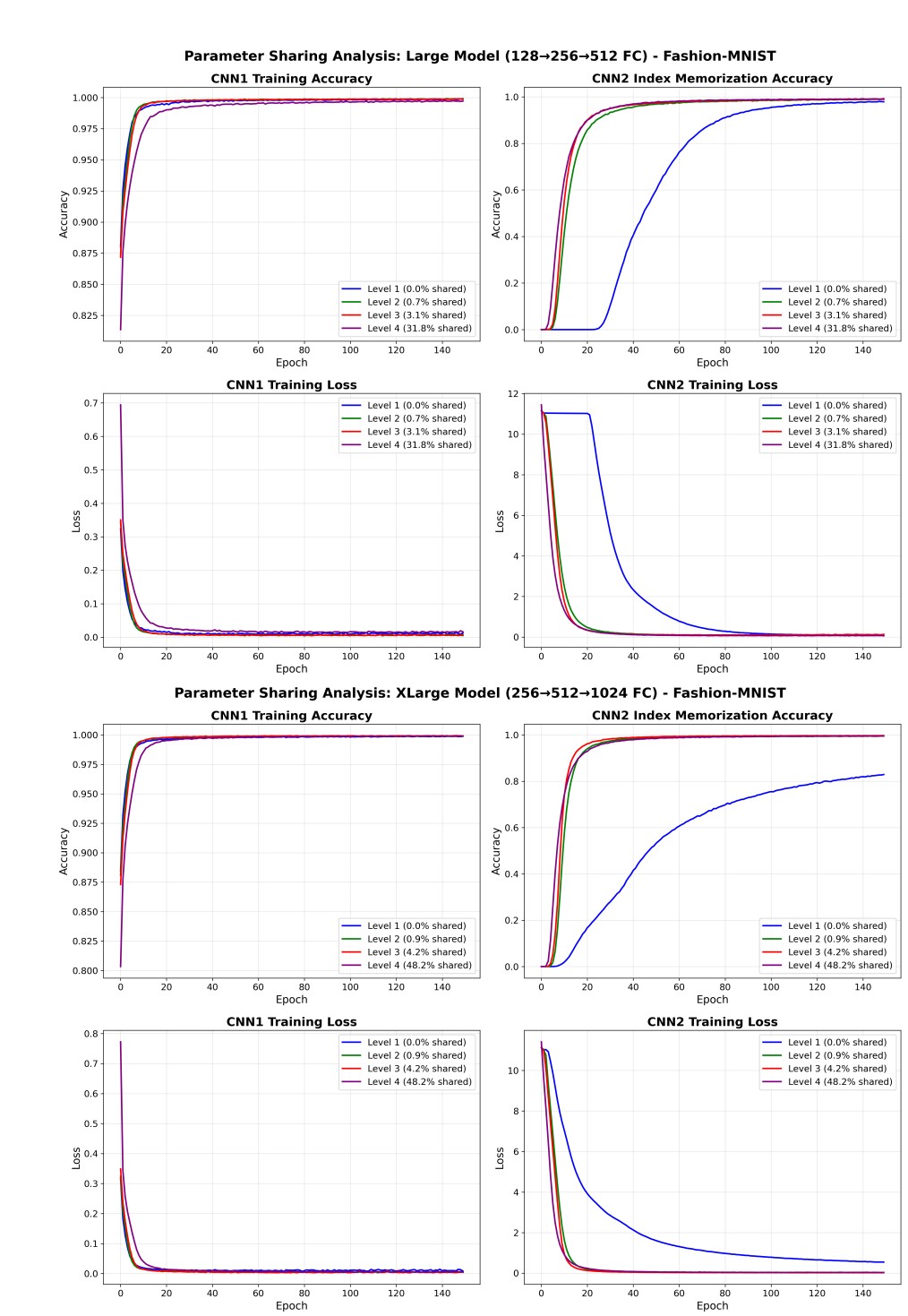

Figure 17: Convergence plots for class branch (CNN1) and index branch (CNN2) for a large (top) and xlarge (bottom) size CNNs over FashionMNIST dataset.

### 7.6.1 TASKS AND TRAINING PROTOCOL

We trained a Two-Staged provenance Network (non-class conditional) for this task.

Batch sizes varied by model scale for memory efficiency. Small, Medium, and Large models used batch sizes of 64 for CNN1 and 32 for CNN2. The XLarge model required reduced batch sizes of 32 and 16 respectively to fit in GPU memory. Learning rates were adjusted based on model capacity: Small and Medium models used 0.001 for CNN1 and 0.0001 for CNN2; Large models used 0.0005 for both networks; XLarge models used 0.0002 for both networks to ensure stable training at massive scale.

### 7.6.2 EVALUATION METRICS

We evaluated models using 4 key metrics as diplayed in Tables 5, 6. CNN1 test accuracy measures classification performance on the held-out test set, indicating generalization capability. CNN2 training accuracy measures index memorization performance on training data, showing the network's capacity to memorize individual instances. CNN2 class consistency (Top-1 and Top-5) evaluates whether memorized training indices preserve semantic structure: when shown a test image, do the top-1 or top-5 predicted training sample indices belong to the correct class? This metric reveals whether memorization captures class-level patterns beyond pure instance recall. Finally, we computed the sharing ratio as the percentage of shared parameters relative to total unique parameters across both networks.

### 7.6.3 CAPACITY AND SHARING DYNAMICS

Model capacity fundamentally shapes parameter sharing dynamics. Small models showed the strongest interference effects, particularly at Level 4 sharing where limited capacity forced direct competition between tasks. Medium and Large models demonstrated that increased capacity reduces interference, enabling near-perfect performance on both tasks even with substantial sharing. The XLarge Model revealed that massive capacity (approximately 80M parameters) can accommodate up to 48% parameter sharing with minimal degradation, suggesting that capacity-constrained interference diminishes as models scale.

Dataset complexity interacted with model capacity in predictable ways. MNIST's simpler visual patterns allowed even Small models to achieve strong performance across sharing levels. Fashion-MNIST's increased complexity revealed clearer capacity constraints: Small models showed significant memorization degradation at high sharing levels, while larger models maintained strong performance. This pattern suggests that complex datasets require proportionally more capacity to support parameter sharing without interference.

### 7.6.4 OPTIMAL SHARING LEVELS

Level 3 sharing (all convolutional layers) emerged as optimal for most configurations, particularly on complex datasets. This level provided sufficient shared feature extraction while preserving task-specific capacity in FC layers. Level 4 sharing (including first FC layer) created the highest parameter overlap but showed performance degradation in capacity-constrained settings, especially for Small models on Fashion-MNIST.

Interestingly, Level 1 sharing (first conv layer only) sometimes underperformed on Fashion-MNIST, particularly for Medium and XLarge models. This suggests that minimal sharing provides insufficient feature extraction capacity for complex visual tasks, and that intermediate sharing levels enable better learned representations through multi-task pressure on shared parameters.

### 7.6.5 MEMORIZATION AND SEMANTIC STRUCTURE

Class consistency metrics revealed that memorization preserves semantic structure beyond pure instance recall. Top-5 class consistency substantially exceeded Top-1 across all configurations, indicating that memorized indices cluster by class even when exact matches are imperfect. This demonstrates that the memorization task implicitly learns class-level representations.

Higher sharing levels generally improved class consistency, suggesting that shared representations encode semantic information more effectively than task-specific features. This pattern was most pronounced in larger models, where Level 3-4 sharing achieved the highest class consistency despite having the greatest parameter overlap. This finding suggests that forcing networks to share representations encourages learning of generalizable semantic features.

### 7.6.6 TRAINING DYNAMICS

Analysis of training curves revealed distinct convergence patterns. CNN1 classification typically plateaued within 20 epochs, indicating rapid learning of discriminative features. CNN2 memorization exhibited slower, more gradual improvement throughout the 150-epoch training period, reflecting the difficulty of learning 60,000-way classification.

Level 1-3 sharing produced smooth, stable loss curves across all model sizes. Level 4 sharing introduced instability in Small models, manifested as oscillating training loss, particularly on Fashion-MNIST. This instability disappeared in larger models, confirming that capacity constraints drive interference effects at high sharing levels.

Higher sharing levels accelerated CNN2 convergence in larger models, suggesting that shared task-relevant features bootstrap memorization learning. This effect was absent in Small models, where capacity constraints prevented efficient feature sharing.

### 7.6.7 IMPLICATIONS

The results challenge the assumption that dramatically different tasks necessarily require separate parameters, showing instead that capacity and sharing level can be tuned to achieve strong multi-task performance.

The finding that Level 3 sharing (all conv layers) often outperforms minimal sharing suggests that multi-task learning pressure improves shared representations. This has practical implications for model design: deliberately sharing mid-level features may produce better representations than keeping networks entirely separate.

The class consistency results reveal that memorization tasks implicitly learn semantic structure, even when trained only on instance-level labels. This suggests that instance-level supervision may be a viable alternative to explicit class labels for learning discriminative representations, particularly in scenarios where class labels are expensive or ambiguous.

Finally, the scaling behavior demonstrates that interference effects diminish with capacity, but not uniformly. The non-monotonic relationship between sharing level and performance (with Level 1 sometimes underperforming Levels 2-3) indicates that sharing dynamics are complex and capacity-dependent, warranting further investigation into optimal parameter sharing strategies across different scales and task combinations.

## 7.7 MEMBERSHIP INFERENCE ANALYSIS SETUP

### 7.7.1 TRAINING CONFIGURATION

We adopt a controlled training setup for membership inference analysis. The key hyperparameters are summarized in Table 7.

Table 7: Training configuration for membership inference analysis.

| Setting | Value |
|---|---|
| Optimizer | AdamW (lr = $2 \times 10^{-3}$, weight decay = $2 \times 10^{-5}$, betas = (0.9, 0.999)) |
| Schedule | Warmup (3 epochs) + Step decay ($\gamma = 0.6$, step = 8 epochs) |
| Epochs | 40 |
| Batch size | 128 |
| Mixed precision | Enabled (AMP) |
| Loss weights | $\alpha = 0.3$ (digit), $\beta = 0.7$ (index) |
| Other | Seed = 42, 4 workers, 5000 MIA samples |
| Evaluation | Top-$k$ provenance accuracy with $k \in \{1, 5\}$ |

### 7.7.2 MODEL: TWO-STAGE CNN ATTRIBUTION

The proposed **Two-Stage CNN Attribution** model jointly predicts class labels and training indices. It consists of:

- **Shared feature extractor:** Three convolutional blocks with BatchNorm, ReLU, and pooling, followed by adaptive pooling to a $4 \times 4$ grid.
- **Projection layer:** Fully connected projection to a 2048-d representation (ReLU, BatchNorm, Dropout).
- **Digit head (class prediction):** A two-layer MLP mapping to the number of classes.
- **Instance head (index prediction):** Conditioned on both the projected features and the class label. During training, the ground-truth label is used; otherwise, the predicted label (argmax) is used. The label is encoded as a one-hot vector and concatenated with the feature representation.

This design ensures that index attribution is conditioned on class identity, mimicking provenance behavior.

### 7.7.3 LOSS FUNCTIONS

We use two objectives:

- **Digit loss:** Standard cross-entropy on class prediction.
- **Instance loss:** Cross-entropy on index prediction with label smoothing (0.05).

The final training objective is a weighted sum:

$$\mathcal{L} = \alpha \, \mathcal{L}_{\text{digit}} + \beta \, \mathcal{L}_{\text{index}}.$$

### 7.7.4 MEMBERSHIP INFERENCE PROTOCOL

For membership inference, we query the index prediction head with candidate samples. Models trained on provenance information tend to assign higher confidence to training members than to non-members. We quantify this effect on held-out data, using 5,000 candidate samples.

Distribution of max confidence scores and entropies over train/member and test/non-member data-point are shown next.

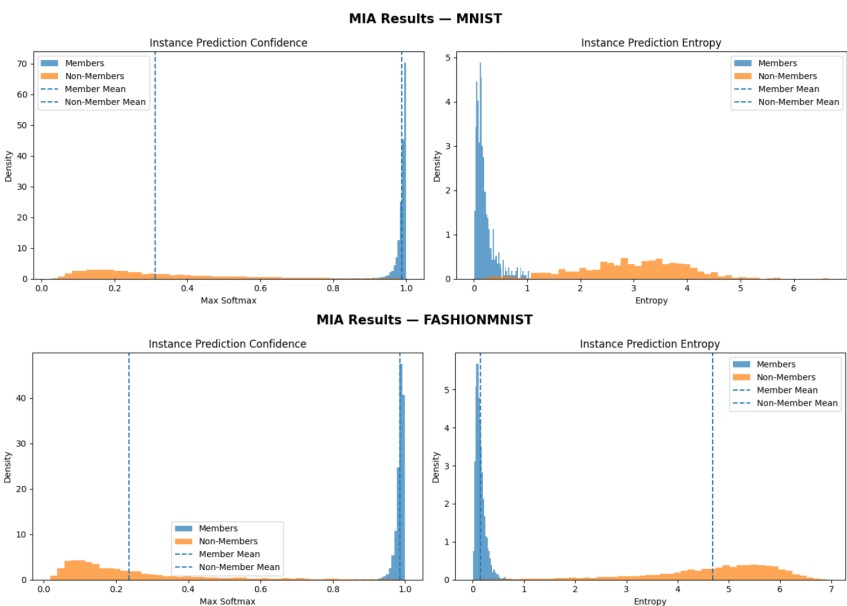

Figure 18: The left panel displays the distribution of maximum confidence scores, while the right panel shows the distribution of entropy, all from the **index branch** of a class-conditional two-branch network. Both distributions are plotted for 5K training samples (members) and 5K test samples (non-members). This data is used to generate the plot in Figure 7
.

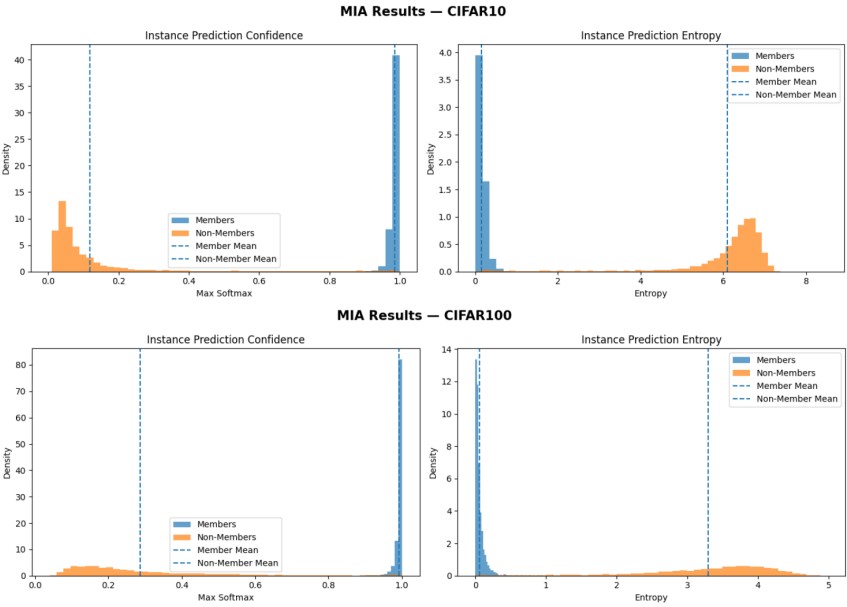

Figure 19: Same as above over CIFAR datasets.

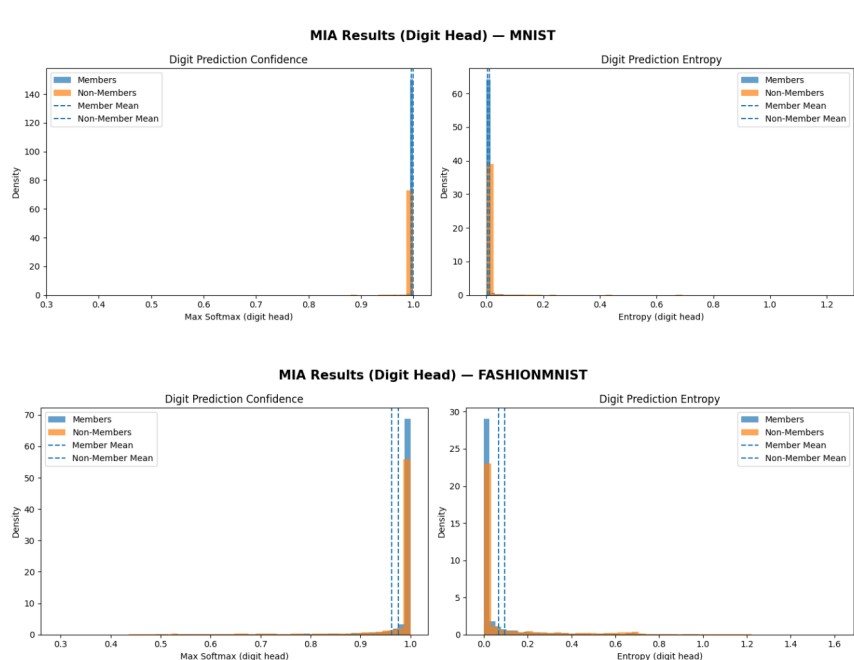

Figure 20: The left panel displays the distribution of maximum confidence scores, while the right panel shows the distribution of entropy, all from the **class branch** of a class-conditional two-branch network. Both distributions are plotted for 5K training samples (members) and 5K test samples (non-members).

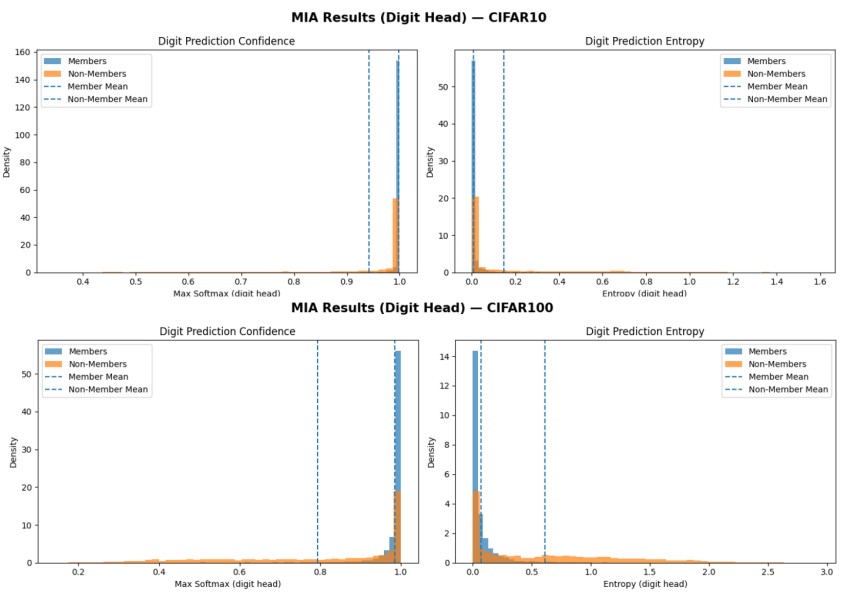

Figure 21: Same as above over CIFAR datasets.

## 7.8 ROBUSTNESS ANALYSIS

We evaluate robustness under a diverse set of input distortions, shown in Figure 24 over four samples. Results over all 14 distortion types are shown in Figure 23.

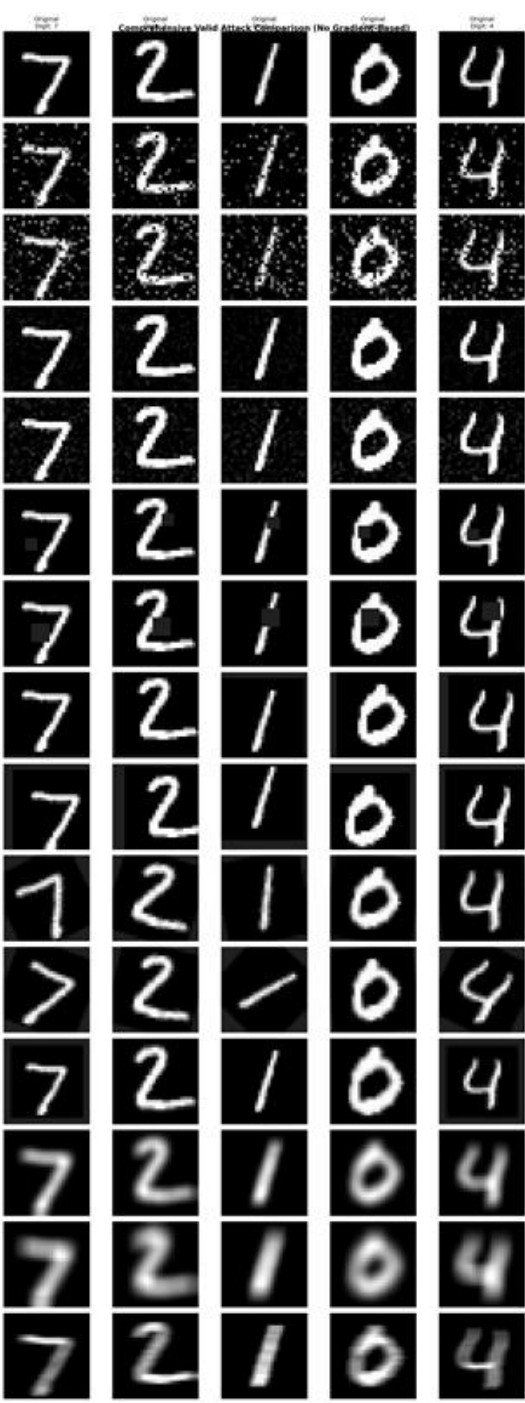

Figure 22: Distortion types in order from top to bottom: no distortion, salt-and-pepper noise (p=0.1, p=0.2), Gaussian noise ($\sigma = 0.1, \sigma = 0.2$), occlusion ($4 \times 4$, $6 \times 6$), translation ($\pm$ 3px, $\pm$ 4px), rotation ($\pm 30°, \pm 40°$), scaling (0.8-1.2x), Gaussian blur ($\sigma = 1, \sigma = 2$), and motion blur (5px).

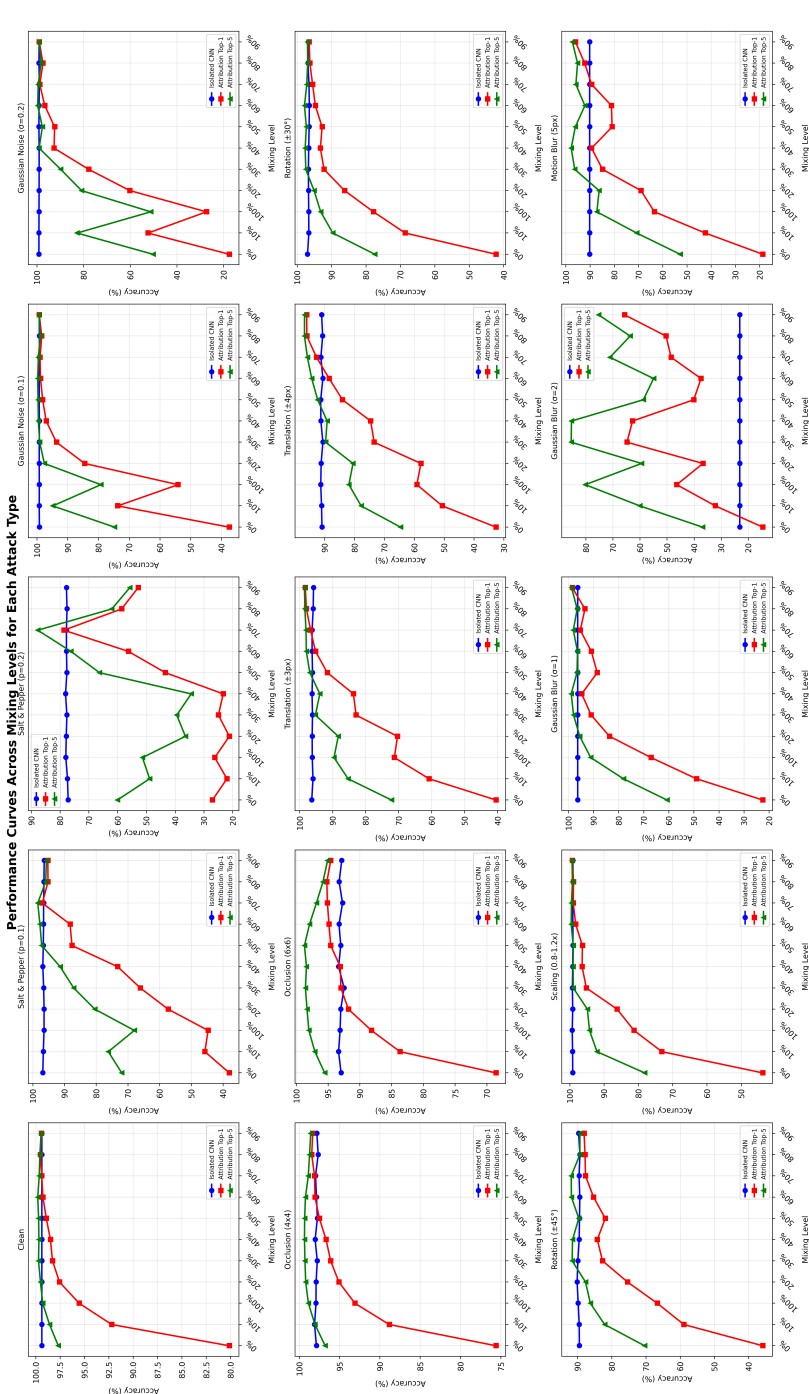

Figure 23: Comparison of the single-branch index-prediction network with varying levels of label mixing against an isolated CNN. Plots show Top-1 and Top-5 accuracy under 14 distortion types plus a baseline without distortion. The variation in the isolated CNN (blue curves) across different index-mixing levels arises from the use of different test sets at each level. Intermediate levels of memorization improve robustness: for distortions like occlusion and blur, partial label mixing (20–30%) yields higher accuracy than the isolated CNN.

## 7.9 DATASET DEBUGGING

Figure 24 shows representative (left) and anomalous (right) training samples, ranked by the entropy of the index-branch output in the two-branch class-conditional network. Low entropy corresponds to sparse neuron activations. The top rows display MNIST samples, while the bottom rows show FashionMNIST samples.

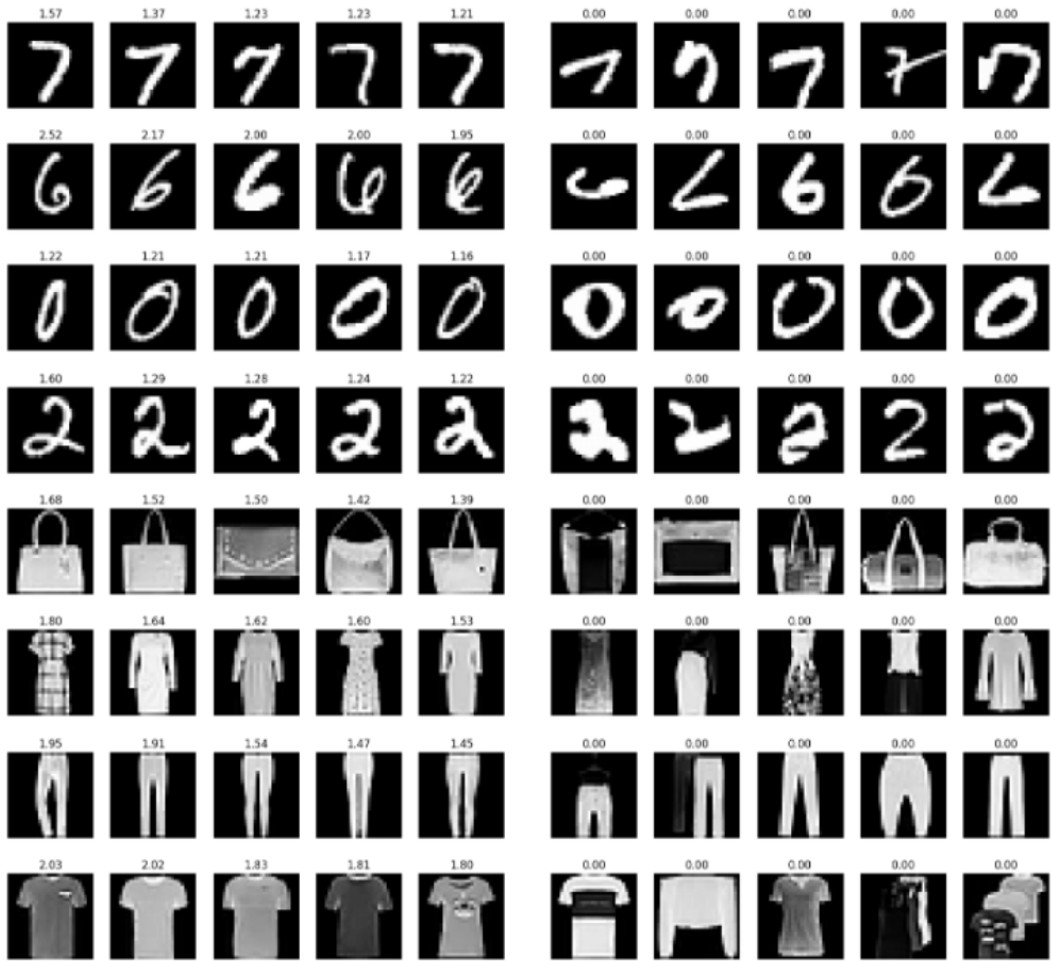

Figure 24: Representative (left) and anomalous (right) training samples ranked by index-branch entropy for the two-branch class-conditional network. Low entropy indicates sparse neuron activation. Top rows: MNIST; bottom rows: FashionMNIST.

## 7.10   IMAGE GENERATION

Our model is a **Variational Autoencoder (VAE)** with an integrated classification module designed to jointly optimize for high-quality generative modeling and discriminative representation learning. The model has three main components: an **encoder**, a **latent reparameterization block**, and a **decoder with auxiliary heads**. Below, we describe each in detail.

### 7.10.1   ENCODER

The encoder $f_\phi : \mathbb{R}^{1 \times 28 \times 28} \to \mathbb{R}^{2d}$ maps the input $x$ to the parameters of a Gaussian distribution in a $d$-dimensional latent space (we use $d = 128$).

The encoder is implemented as a three-stage convolutional feature extractor followed by two parallel fully connected heads:

- **Conv Stage 1:** $3 \times 3$ convolution with $64$ output channels, stride 2, and padding $1 \to$ BatchNorm $\to$ ReLU. Reduces spatial resolution from $28 \times 28$ to $14 \times 14$.

- **Conv Stage 2:** $3 \times 3$ convolution with $128$ channels, stride 2, padding $1 \to$ BatchNorm $\to$ ReLU. Further reduces resolution to $7 \times 7$.

- **Conv Stage 3:** $3 \times 3$ convolution with $256$ channels, stride 1, padding $1 \to$ BatchNorm $\to$ ReLU. Maintains $7 \times 7$ spatial size while enriching representation depth.

The resulting $256 \times 7 \times 7$ feature map is flattened into a vector of size $12544$ and passed through two linear layers:

$$\mu(x) = W_\mu h + b_\mu, \qquad \log \sigma^2(x) = W_{\log \sigma} h + b_{\log \sigma}.$$

### 7.10.2   LATENT REPARAMETERIZATION

We use the reparameterization trick to sample $z \sim q_\phi(z|x)$:

$$z = \mu(x) + \sigma(x) \odot \epsilon, \qquad \epsilon \sim \mathcal{N}(0, I).$$

This allows gradients to propagate through stochastic sampling during training, enabling end-to-end optimization of both encoder and decoder parameters.

## 7.11   DECODER

The decoder $g_\theta : \mathbb{R}^d \to \mathbb{R}^{1 \times 28 \times 28}$ reconstructs the input image from $z$.

- **Fully Connected Projection:** The latent code is mapped back to a $256 \times 7 \times 7$ tensor.

- **Deconv Stage 1:** $3 \times 3$ transposed convolution to $128$ channels, stride $1 \to$ BatchNorm $\to$ ReLU. (Resolution remains $7 \times 7$.)

- **Feature Pooling for Prediction:** We perform global average pooling over this intermediate 128-channel feature map, yielding $h_{\text{cls}} \in \mathbb{R}^{128}$. This vector is used for auxiliary prediction heads:

  - **Class Head:** A linear layer producing logits for $K$ classes (we use $K = 10$).
  - **Index Heads (optional):** A list of linear layers, each predicting a separate categorical factor.

- **Deconv Stage 2:** $3 \times 3$ transposed convolution to $64$ channels, stride 2, padding 1, output padding $1 \to$ BatchNorm $\to$ ReLU. (Upsamples to $14 \times 14$.)

- **Deconv Stage 3:** $3 \times 3$ transposed convolution to $32$ channels, stride 2, padding 1, output padding $1 \to$ BatchNorm $\to$ ReLU. (Restores full $28 \times 28$ resolution.)

- **Output Layer:** $3 \times 3$ convolution producing a single channel, followed by a sigmoid nonlinearity to ensure pixel intensities lie in $[0, 1]$.

### 7.11.1 TRAINING SETUP

We train the model on the **IndexedMNIST** dataset, which augments the standard MNIST digits with class indices for auxiliary prediction tasks. Input images are normalized to $[0, 1]$. We use a batch size of $128$ for training and $64$ for evaluation.

### 7.11.2 OPTIMIZATION

The model parameters are optimized using the Adam optimizer with learning rate $10^{-3}$ for $70$ epochs. For each minibatch, we compute:

- **Reconstruction loss:** Binary cross-entropy between input $x$ and reconstruction $\hat{x}$.
- **KL divergence:** Regularizing the approximate posterior $q_\phi(z|x)$ toward the prior $p(z) = \mathcal{N}(0, I)$.
- **Classification loss:** Cross-entropy over the class logits.
- **Index loss (optional):** Cross-entropy over each auxiliary index head.

The total training objective is:

$$\mathcal{L} = \lambda_{\text{gen}} \Big( \underbrace{\mathbb{E}_{q_\phi(z|x)}[-\log p_\theta(x|z)]}_{\text{Reconstruction Loss}} + \underbrace{D_{\text{KL}}(q_\phi(z|x) \,\|\, p(z))}_{\text{KL Regularization}} \Big) + \lambda_{\text{cls}} \Big( \underbrace{\mathcal{L}_{\text{CE}}(y, \hat{y})}_{\text{Classification Loss}} + \underbrace{\sum_k \mathcal{L}_{\text{CE}}(y_k, \hat{y}_k)}_{\text{Index Losses}} \Big).$$

During training, we log the average total loss, its decomposition (reconstruction, KL, classification, index), and the classification and index prediction accuracies. This provides a clear picture of generative quality and discriminative performance over time.

PyTorch implementations are provided next.

```python
class Encoder(nn.Module):
    def __init__(self, latent_dim=64):
        super().__init__()
        self.enc = nn.Sequential(
            nn.Conv2d(1, 64, 3, stride=2, padding=1),
            nn.BatchNorm2d(64),
            nn.ReLU(),
            nn.Conv2d(64, 128, 3, stride=2, padding=1),
            nn.BatchNorm2d(128),
            nn.ReLU(),
            nn.Conv2d(128, 256, 3, stride=1, padding=1),
            nn.BatchNorm2d(256),
            nn.ReLU(),
        )
        self.fc_mu = nn.Linear(256*7*7, latent_dim)
        self.fc_logvar = nn.Linear(256*7*7, latent_dim)

    def forward(self, x):
        x = self.enc(x)
        x = x.view(x.size(0), -1)
        mu = self.fc_mu(x)
        logvar = self.fc_logvar(x)
        return mu, logvar
```

Figure 25: Encoder of VAE on MNIST and FashionMNIST datasets

```python
class Decoder(nn.Module):
    def __init__(self, latent_dim=64):
        super().__init__()
        self.fc = nn.Linear(latent_dim, 256*7*7)
        self.up1 = nn.ConvTranspose2d(256, 128, 3, stride=1, padding=1)
        self.bn1 = nn.BatchNorm2d(128)
        self.up2 = nn.ConvTranspose2d(128, 64, 3, stride=2, padding=1, output_padding=1)
        self.bn2 = nn.BatchNorm2d(64)
        self.up3 = nn.ConvTranspose2d(64, 32, 3, stride=2, padding=1, output_padding=1)
        self.bn3 = nn.BatchNorm2d(32)
        self.outc = nn.Conv2d(32, 1, 3, padding=1)

        self.pool = nn.AdaptiveAvgPool2d(1)
        self.class_head = nn.Linear(128, 10)
        self.index_heads = None

    def forward(self, z):
        x = self.fc(z).view(-1, 256, 7, 7)
        x = F.relu(self.bn1(self.up1(x)))
        h_cls = self.pool(x).flatten(1)
        class_logits = self.class_head(h_cls)
        index_logits = None
        if self.index_heads is not None:
            index_logits = [head(h_cls) for head in self.index_heads]
        x = F.relu(self.bn2(self.up2(x)))
        x = F.relu(self.bn3(self.up3(x)))
        recon = torch.sigmoid(self.outc(x))
        return recon, class_logits, index_logits
```

Figure 26: Decoder of VAE on MNIST and FashionMNIST datasets

```python
class VAE(nn.Module):
    def __init__(self, latent_dim=64, class_sizes=None):
        super().__init__()
        self.encoder = Encoder(latent_dim)
        self.decoder = Decoder(latent_dim)
        if class_sizes is not None:
            self.decoder.index_heads = nn.ModuleList([nn.Linear(128, sz) for sz in class_sizes])
        self.latent_dim = latent_dim

    def reparameterize(self, mu, logvar):
        std = torch.exp(0.5*logvar)
        eps = torch.randn_like(std)
        return mu + eps*std

    def forward(self, x):
        mu, logvar = self.encoder(x)
        z = self.reparameterize(mu, logvar)
        recon, class_logits, index_logits = self.decoder(z)
        return recon, mu, logvar, class_logits, index_logits
```

Figure 27: VAE Architecture on MNIST and FashionMNIST datasets

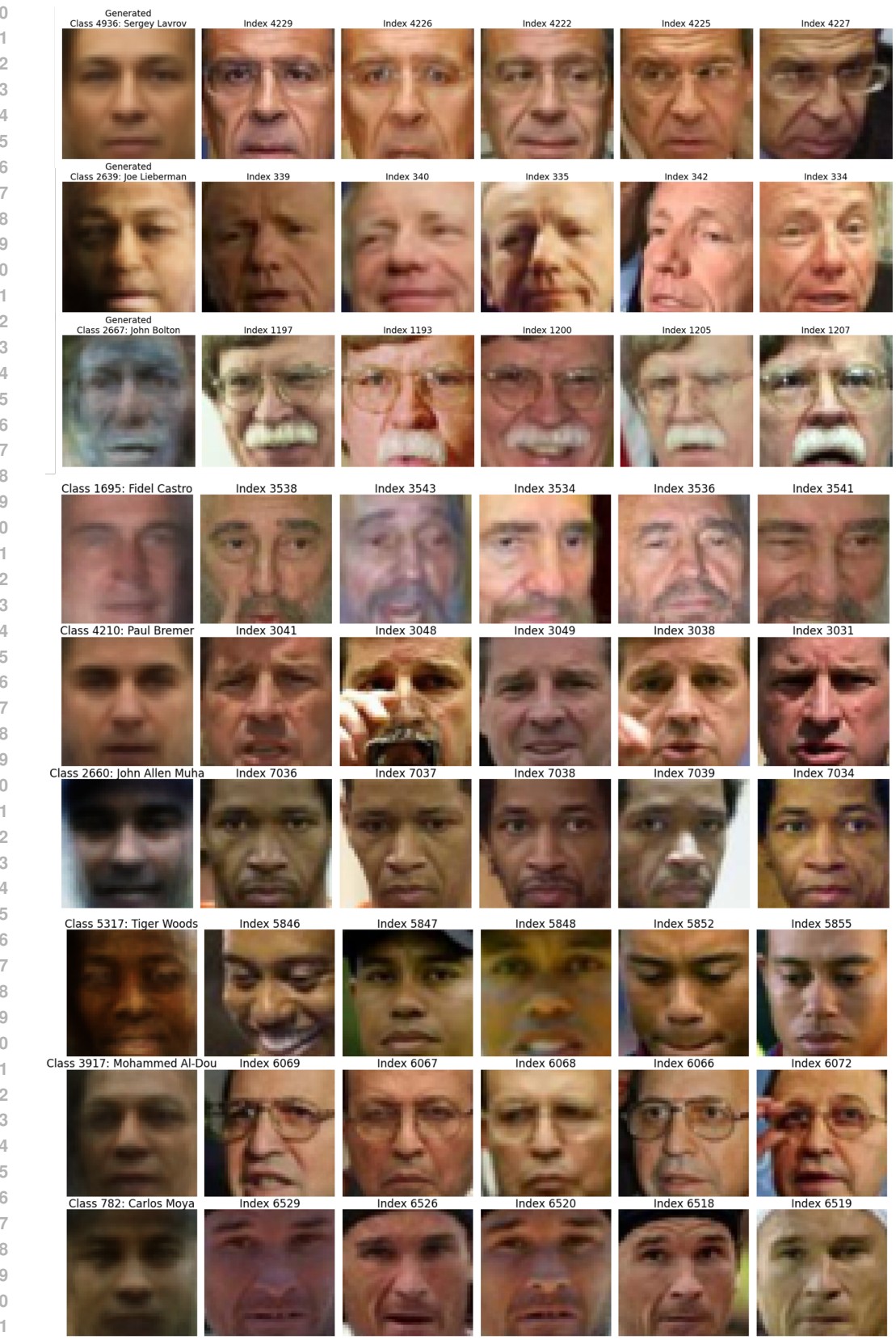

Figure 28: Additional generated faces by VAE (left column) along with five most similar training faces derived from the class and index branches.

### 7.11.3 ANALYZING THE RELATIONSHIP BETWEEN GENERATED IMAGE REALISM AND THE INDEX-PREDICTION BRANCH CONFIDENCE

We assessed the relationship between generated image realism and the index-prediction branch confidence. Specifically, we split generated samples by confidence (conf $< 0.5$ vs conf $\geq 0.5$). To quantify the continuous association we report Spearman's rho between sample-level confidence and an individual-image quality proxy (Inception features distance to nearest training sample / reconstruction error). Across four runs, images from the high-confidence group had substantially lower FID than the low-confidence group:

| Threshold | $N_{high}$ | $N_{low}$ | FID$_{high}$ (95% CI) | FID$_{low}$ (95% CI) | Spearman $\rho$ (p) |
|---|---|---|---|---|---|
| 0.5 | 432 | 568 | 28.7 (25.6–31.8) | 65.4 (61.1–69.9) | $-0.62$ ($< 10^{-6}$) |

Results show a consistent negative association: higher-confidence samples have substantially lower FID and smaller feature-space distances.

## 7.12 INFLUENCE ESTIMATION VIA LAST-LAYER REPRESENTATIONS

Understanding which training examples are responsible for a model's prediction has become a central problem in interpretability. Classical influence functions estimate the effect of removing a training point $(x_i, y_i)$ on the loss at a test point $x_t$ via a second-order Taylor approximation of the ERM objective (Koh & Liang, 2017b). However, computing influence scores requires Hessian–vector products and often suffers from numerical instability in deep networks (Basu et al., 2020; Feldman, 2020). This makes traditional influence estimation extremely expensive and frequently unreliable in practice.

In contrast, recent theoretical and empirical findings show that for deep networks with a linear final layer—which includes essentially all modern CNNs and transformers—**the influence of a training example can be well-approximated by nearest neighbors in the last-layer representation space** (Yeh et al., 2018; Pruthi et al., 2020). Let $f_\theta : \mathcal{X} \to \mathbb{R}^d$ denote the network embedding (all layers except the final classifier). The logit for class $c$ is then

$$F_c(x) = w_c^\top f_\theta(x), \tag{22}$$

where $w_c \in \mathbb{R}^d$ are the classifier weights.

**Influence as Representation Similarity.** For a training example $(x_i, y_i)$ define its embedding $z_i = f_\theta(x_i)$ and similarly $z_t = f_\theta(x_t)$ for a test example. Under mild regularity assumptions, the first-order approximation of the change in logits when replacing $x_t$ with $x_i$ in training yields an influence score proportional to the similarity of their representations:

$$I(i \to t) \ \propto \ \langle z_i, z_t \rangle. \tag{23}$$

Cosine similarity provides a numerically stable version:

$$S(i, t) = \frac{\langle z_i, z_t \rangle}{\|z_i\|_2 \, \|z_t\|_2}. \tag{24}$$

Thus, the **most influential training example** is simply the nearest neighbor of $x_t$ in representation space:

$$i^* = \arg \max_{i \in \mathcal{D}_{\text{train}}} S(i, t). \tag{25}$$

Assigning the label of the most influential training point yields the influence-based prediction:

$$\hat{y}_t = y_{i^*}. \tag{26}$$

**Representer Theorem Connection.** Yeh et al. (2018) show that for deep networks trained with weight decay, each prediction can be decomposed as a weighted sum of training example similarities:

$$F_c(x_t) = \sum_{i=1}^{n} \alpha_{i,c} \langle z_i, z_t \rangle, \tag{27}$$

where $\alpha_{i,c}$ depend on the classifier weights and loss. In many classification settings, the coefficients $\alpha_{i,c}$ are approximately uniform within a class, yielding a natural justification for nearest-neighbor influence estimation.

**Practical Advantages.** The last-layer influence approximation offers:

- **No gradients or Hessians**: only forward passes.
- **Low memory**: store $d$-dim embeddings, not full gradients.
- **Stability**: representation similarity is smooth and robust.
- **Faithfulness**: provably consistent with linear last layers.

Empirically, on MNIST and CIFAR-10, the influence-NN classifier achieves accuracy close to classical nearest neighbors in the learned embedding space, far exceeding gradient-based influence methods which often perform near chance level due to noise.

This approach therefore serves as an inexpensive and theoretically grounded proxy for influence functions, while avoiding the instability, memory cost, and high computational burden of gradient-based methods.

The code is available at [MASKED].

**Algorithm 1** Influence-Based Prediction via Last-Layer Nearest Neighbors

**Require:** Trained network $f_\theta$, training set $\{x_i, y_i\}$, test input $x_t$
1: Precompute embeddings $z_i = f_\theta(x_i)$ for all training points.
2: Compute test embedding $z_t = f_\theta(x_t)$.
3: Compute similarities $S(i, t)$ for all $i$:

$$S(i, t) = \frac{\langle z_i, z_t \rangle}{\|z_i\|_2 \|z_t\|_2}.$$

4: Identify the most influential example:

$$i^* = \arg\max_i S(i, t).$$

5: **return** predicted label $\hat{y}_t = y_{i^*}$.

