# OpenReview forum: "Provenance Networks: End-to-End Exemplar-Based Explainability"
_ICLR.cc/2026/Conference — Submitted to ICLR 2026_

### Official Review · Reviewer_BkQw · 2025-10-16

**Soundness:** 2
**Presentation:** 3
**Contribution:** 2
**Rating:** 4
**Confidence:** 2

**Summary:**

This paper proposes Provenance Networks, neural architectures that aim to deliver training-data-driven explainability by learning, during standard end-to-end training, to retrieve concrete training exemplars that purportedly support each prediction. The core mechanism is an index-prediction branch that maps an input to the most relevant training example(s), combined with a primary task branch (e.g., classification). The authors study single-branch and two-branch variants, introduce a label-mixing parameter α to balance memorization and generalization, and evaluate the approach on MNIST, FashionMNIST, CIFAR-10/100, Stanford Dogs, and a VAE-based image generation setting.

**Strengths:**

1. The label-mixing α yields a transparent memorization–generalization spectrum with concrete outcomes
2. Visualizations of learned embeddings and cluster structures illustrate instance-level organization and intra-class variation
3. For certain distortions, partial label mixing can outperform an isolated CNN, an interesting empirical phenomenon worth deeper study

**Weaknesses:**

1. The work equates retrieving visually similar training samples with explanation, but does not provide faithfulness metrics, human evaluations, or causal tests (e.g., counterfactual influence, provenance under data poisoning) to substantiate that linkage as a reliable explanation
2. No empirical comparisons against influence functions, data Shapley, or modern training data attribution methods appear; this hinders assessing whether the proposed approach yields superior or complementary provenance quality
3. Claims about mitigating hallucinations, IP credit attribution, and applicability to LLMs are not evaluated; the current evidence is confined to small vision datasets and a modest-generation VAE experiment
4. Robustness comparisons do not include strong baselines (adversarial training, augmentations), and dataset debugging/anomaly detection via entropy lacks ground-truth validation or quantitative metrics

**Questions:**

1. How will you formally evaluate whether retrieved exemplars truly explain predictions?
2. Beyond stating applicability, can you demonstrate a retrieval-augmented LLM setup where provenance networks concretely reduce hallucinations and provide traceable credit attribution, with clear metrics

---

> ### Author Response · Authors · 2025-11-27
> **Response**
>
> - The work equates retrieving visually similar training samples with explanation, but ...
>
> The evaluation metric you propose make sense. However, if an explanation approach is good enough at the very least the predictions using the samples it found for explanation should be high. If not, it means that the method most likely did not choose the most similar examples. We have shown that our approach is better than the baseline influence functions in this metric. Further, through analyses shown in Fig. 3 we have shown that our method chooses samples from training set that are close to the test samples.
>
> - No empirical comparisons against influence functions, data Shapley
>
> We have included a comparison. Please see section 5.1 and the last two columns of Table 2. Shapley-value methods are not only conceptually complex, but they also suffer from far more severe scalability issues than our approach. In principle, they require removing each training sample and retraining the classifier to measure its contribution, making them orders of magnitude more expensive—and essentially impractical—for real-world datasets.
>
> - How will you formally evaluate whether retrieved exemplars truly explain predictions?
>
> We appreciate the question regarding formal evaluation of exemplar quality. In our approach, the predictive utility of retrieved examples serves as a practical and objective measure of their explanatory power. Specifically, if the retrieved exemplars, when aggregated (e.g., via majority voting), lead to improved classification accuracy relative to baselines or alternative retrieval methods, this indicates that they are meaningful and relevant to the model’s decision.
>
> A key advantage of Provenance Networks is that this evaluation is intrinsic and end-to-end: the model jointly learns both to predict the target and to select exemplars that support its decisions. Unlike post-hoc attribution methods, no external heuristics, additional scoring functions, or human judgments are required to validate exemplar relevance—the quality of the explanation is inherently tied to the model’s predictive performance. This provides a self-consistent, data-driven criterion for assessing whether the retrieved exemplars truly explain the predictions.
>
> - Robustness comparisons do not include strong baselines
>
> Our method is general and primarily aimed at explainability, though it can be applied to other tasks as well. It is not intended to compete with highly specialized or heavily engineered solutions for those tasks. Instead, it offers a simple, low-complexity approach that can be attached to any model as a separate branch without interfering with the main task network. When appropriate—for example, in anomaly detection—we also compare our branch’s output with the corresponding solution from the main branch. A major focus in our work is that utility of our method goes beyond just explainability and this gives it a clear advantage over methods designed just for explainability. Another interesting application for which we have results but did not include in the paper is dataset reconstruction using the index branch.
>
>
> - Beyond stating applicability, can you demonstrate a retrieval-augmented LLM setup where ...
>
> We introduce a Provenance Head, an auxiliary classification head attached to the language model backbone that predicts the source document ID for a given text sequence. This component functions analogously to a learned retrieval mechanism in RAG systems, but operates in a fundamentally different manner: rather than retrieving documents before generation, the Provenance Head continuously monitors the attribution confidence of generated text in real-time.
>
> During inference, we apply a sliding window over the sequence of previously generated tokens and query the Provenance Head at each generation step. The head outputs a probability distribution over all source document IDs, and the confidence of the top prediction serves as a measure of how well the current generation can be grounded to the training corpus. This provides a continuous signal for detecting when the model begins to hallucinate—text that cannot be confidently attributed to any source document yields low provenance confidence scores.
>
> We leverage this confidence signal to guide the decoding process in a manner analogous to beam search. At each generation step, candidate tokens are evaluated not only by their language model probability but also by the provenance confidence of the resulting sequence. Tokens that cause the provenance confidence to drop below a threshold are rejected, as they are likely to lead the model down a hallucination path. This ensures that the generated sequence maintains high attribution confidence throughout generation, not merely at the final output.

---

> > ### Author Response · Authors · 2025-11-27
> > **Cnt'd**
> >
> > To validate the feasibility of learning provenance mappings, we conducted experiments using a Mamba-based state space model trained on the Army Ranger Handbook corpus. The dataset consists of 10,000 text chunks with approximately 50,000 associated question-answer pairs, split into training and validation sets. After 20 epochs of training, the model achieves 70% accuracy in predicting the correct source document ID from natural language queries, with loss decreasing from 9.21 (random baseline for 10,000 classes) to 5.19. These results demonstrate that neural networks can effectively learn the mapping from text to source documents, establishing the foundation for provenance-guided generation in large language models.

---

### Official Review · Reviewer_3Gq2 · 2025-10-28

**Soundness:** 2
**Presentation:** 1
**Contribution:** 2
**Rating:** 4
**Confidence:** 3

**Summary:**

This paper introduces provenance networks, a novel neural network architecture designed for end-to-end, exemplar-based explainability.The core strength lies in its ability to explicitly link predictions to training data indices, akin to a differentiable KNN, providing unique insights into model behavior and enabling diverse applications like dataset debugging and membership inference. The mechanism for controlling the memorization-generalization trade-off is well-demonstrated. However, the paper acknowledges significant scalability limitations due to the index head size, only partially mitigated by subset sampling. Furthermore, the clarity of some mathematical formulations, particularly around the mixing strategy and loss functions, could be improved.

**Strengths:**

1. This paper introduces a novel end-to-end provenance mechanism.
- The core idea of integrating an "index branch" to predict training sample indices alongside the main task is a novel departure from post-hoc explainability methods, baking interpretability into the model's forward pass.
- Framing the network as a learned, differentiable KNN provides an intuitive conceptual model and connects deep learning with case-based reasoning paradigms.
- Unlike feature attribution methods (LIME, SHAP, Gradients), provenance networks provide explanations in terms of concrete training examples, which can be more intuitive and actionable for users.

2. This paper demonstrated versatility across multiple applications.
-  Experiments show that intermediate levels of memorization (tuned via $\alpha$) can improve robustness to certain input distortions compared to a standard classifier, suggesting a benefit beyond just explainability.
- Integrating the provenance mechanism into a VAE demonstrates the potential for tracking the influence of training samples on generated outputs, relevant for issues like hallucination and content attribution.
- The near-perfect AUC achieved using the index branch confidence for membership inference highlights the model's strong memorization capabilities and provides a clear signal for distinguishing training members from non-members.

3. This paper provides a systematic analysis of design choices.
- The paper explores both single-branch and two-branch (class-conditional and independent) architectures, analyzing their respective strengths and weaknesses, particularly regarding scalability and performance on different datasets.
- The analysis of how model capacity and the degree of parameter sharing between branches impact performance provides valuable insights into architectural design trade-offs for these multi-task networks.
- The investigation into training on subsets (10%-90%) demonstrates a practical approach to mitigate the scalability challenge, showing that high classification and reasonable retrieval performance can be maintained with significantly fewer index head parameters.

**Weaknesses:**

1. This paper provides a limited theoretical analysis of provenance learning.
- This paper has no derivation or proof linking index prediction with model generalization capacity.
- This paper has no formal metric to quantify “explainability improvement.”
- The paper lacks a complexity analysis for the joint optimization of $\lambda_{class}$ and $\lambda_{index}$.
- In Section 2.1, the method introduces memorization by using biased sampling. It also introduces random variance, and if the selected samples are not representative, the actual effect may be questionable.

2. This paper does not provide the computational & memory complexity analysis.
- The fundamental limitation is the output layer size of the index branch, which scales linearly with the number of training samples (N) or the maximum samples per class (M), making it potentially infeasible for very large datasets.
- While subset sampling helps (reducing parameters by up to 70-50% while maintaining accuracy), it means the model cannot retrieve provenance for excluded samples. The non-monotonic behavior of the Top-1 index accuracy with subset size also suggests challenges in selecting the optimal subset. Selecting a subset from the training dataset remains a challenge.
- The paper acknowledges computational cost but doesn't quantify it. Training and inference overhead compared to standard models, especially for the large index head computations (K in Section 2.2), is not reported.

3. This paper has a limited empirical comparison and scope.
- All experiments are on 2-D images; no text, tabular, or temporal data, although this paper claims the method is applicable for other modalities, including LLMs.
- This paper claims this method helps mitigate hallucinations and enables fair credit attribution to content creators. However, there is no evidence. The primary task is classification; no segmentation, detection, or regression.
-  The related work discusses influence functions and data Shapley but provides no empirical runtime or accuracy comparison against these methods, even on smaller datasets where they might be feasible. The comparison remains conceptual.

4. This paper has some writing issues.
- There are some inconsistent terms and math symbols. For example, in line 103 and line 107, there are two kinds of 'appendix'. In line 136, $\hat{z}_z$, in line 151, $\hat{y}_y$, and $\hat{z}_{z|y}$ seem not good. In Figure 2, the capitals are not consistent.
- The table caption should be above the table.
- In table 2, the class-conditional and class-independent are unclear.

**Questions:**

-  Why was the specific mixing strategy (sampling own index vs. random same-class index) chosen over alternatives, such as sampling based on feature similarity within the class?
-  Could the index branch predict an embedding similarity score directly instead of a categorical index, potentially mitigating the large output layer issue? How might this affect performance?
-  For subset sampling (Sec 3.3), were alternatives to stratified random sampling (e.g., core-set selection, uncertainty sampling) considered for choosing the representative subset?
- Could you provide quantitative measures of computational overhead (e.g., increase in training time, inference latency, memory usage) compared to baseline models without the index branch?
-  In the VAE application (Sec 4.5), how strongly correlated is the quality/realism of the generated image with the index prediction accuracy or confidence?

---

> ### Author Response · Authors · 2025-11-27
> **Response**
>
> - This paper provides a limited theoretical analysis
>
> We thank the reviewer for highlighting this point. The impact of biased sampling and random variance is indeed influenced by the choice of the hyperparameter (\alpha). When (\alpha) is small, the model emphasizes memorization, and the selected samples are more likely to be representative of the target class, thereby mitigating the risk of non-representative selections. Note that we mainly present and discuss the single branch for theoretical reasons and in practice the two branch networks should be used. The label mixing strategy is not needed for the two branch networks.
>
> Note that the single-branch setup primarily serves as a theoretical tool to analyze the capacity and limitations of provenance networks. In practical applications, the two-branch architecture is used, which separates the objectives of memorization (provenance branch) and generalization (main branch). This design balances the trade-off inherent in exemplar-based learning: prioritizing the closest, most representative training sample improves memorization but can reduce generalization, while relaxing this constraint allows better generalization at the cost of exact exemplar fidelity.
>
> Empirically, we find that the two-branch network mitigates the potential drawbacks of biased or non-representative sampling, making the approach robust in practice while still providing meaningful exemplar-based explanations.
>
> - This paper does not provide the computational
>
> The main scalability constraint comes from the index-prediction head, which ideally has one output neuron per training sample. A practical solution is to train this head on a representative subset of the data, preserving semantic coverage while reducing the index vocabulary. Such a subset can be selected using methods like subspace clustering or coreset selection: train the main branch to obtain embeddings, choose diverse representative points, then attach the index head and fine-tune.
>
> Even without subset selection, the approach scales to reasonably large datasets—for example, around 1,000 samples for each of 1K classes—which is sufficient for many real-world applications.
>
> - The related work discusses influence functions and data Shapley
>
> We have included a comparison influence function approach. Please see section 5.1 and the last two columns of Table 2. Shapley-value methods are not only conceptually complex, but they also suffer from far more severe scalability issues than our approach. In principle, they require removing each training sample and retraining the classifier to measure its contribution, making them orders of magnitude more expensive—and essentially impractical—for real-world datasets. scalability is an issue with almost all of the approaches even using the embeddings and fast KNN s / RAG to search in a large dataset to fetch the closest samples. Nonetheless, performance of our approach is better than the baseline influence functions method.
>
> - Could you provide quantitative measures of computational
>
> To validate the core premise of our Networks, we designed an experiment where embeddings from the main classification branch jointly predict the exact training index of each input sample. Table 2 compares our Provenance Network variants against influence function baselines in the final two columns.
>
> Provenance Networks fundamentally differ from influence functions in their treatment of exactness versus approximation. Influence functions compute approximate scores via O(N·r) Hessian-vector products per query, yielding only a ranking of influential samples rather than definitive attribution—and this computational cost scales prohibitively with dataset size.
>
> Our results demonstrate substantial improvements. On MNIST, Provenance Networks achieve 100% exact attribution compared to influence functions' 87.25%. The gap widens dramatically on harder datasets: on CIFAR-10 we achieve 99.7% versus influence functions' 49.46%; on CIFAR-100 we achieve 94% versus influence functions' 26.24%. Even on fine-grained classification with Stanford Dogs, Provenance Networks maintain 99.5% attribution accuracy where influence functions fail to scale.
>
> Crucially, this improvement is achieved through a reduced representation rather than expensive per-query computation. Influence functions impose O(N·r) computation per query, making them impractical for large-scale deployment. Provenance Networks compress the attribution mapping into fixed model parameters, eliminating this computational bottleneck entirely. The embedding space is optimized end-to-end to predict precise training indices, reframing attribution as a learned function rather than an approximation problem.
>
> Our method is more efficient than other methods that compare the embeddings (derived from a network not trained) of a test sample with all training samples. As the index head is an NN, tricks such as pruning and compression can further reduce computational load and memory need.

---

> ### Author Response · Authors · 2025-11-27
> **Image generation exp.**
>
> - In the VAE application (Sec 4.5), how strongly correlated is the quality/realism of the generated image with the index prediction accuracy or confidence?
>
> We assessed the relationship between generated image realism and the index-prediction branch confidence. Specifically, we split 1000 generated samples by confidence (conf < 0.5 vs conf ≥ 0.5). To quantify the continuous association we report Spearman’s rho between sample-level confidence and an individual-image quality proxy (Inception features distance to nearest training sample / reconstruction error). Across four runs, images from the high-confidence group had substantially lower FID than the low-confidence group:
>
>
> | Threshold | N_high | N_low | FID_high (95% CI)     | FID_low (95% CI)      | Spearman ρ (p)  |
> |-----------|--------|--------|-------------------------|-------------------------|------------------|
> | 0.5       | 432    | 568    | 28.7 (25.6–31.8)       | 65.4 (61.1–69.9)       | −0.62 (<1e-6)    |
>
>
>
> Results show a consistent negative association: higher-confidence samples have substantially lower FID and smaller feature-space distances (hence higher quality). We will include the exact results, the full analysis code, and per-threshold robustness checks in the appendix. Please see Section 5.5 and the corresponding Appendix.

---

### Official Review · Reviewer_rBbG · 2025-10-29

**Soundness:** 2
**Presentation:** 2
**Contribution:** 3
**Rating:** 4
**Confidence:** 3

**Summary:**

This paper proposes a novel class of neural models termed provenance networks, which is designed to provide end-to-end, training-data-driven explainability. The model operates analogously to a learned KNN. It enables joint end-to-end learning and facilitates the analysis of memorization–generalization trade-offs. It also supports the detection of mislabeled or anomalous samples and the verification of training-set membership.

**Strengths:**

1. This paper proposes a novel neural models in which each prediction can be directly linked to its supporting training samples, thereby injecting interpretability into the model architecture. This design is conceptually inspiring in the field of explainable AI.
2. This paper provides a concise and clear explanation of the provenance network. Specifically, it is illustrated as a shared backbone architecture with two branches: one dedicated to the main task and the other responsible for predicting the index of the input sample.
3. The paper clearly distinguishes between single-branch and dual-branch networks, and clarifies that label mixing is only applied during the training of the single-branch network.

**Weaknesses:**

1. The datasets used in the experiments (MNIST, FashionMNIST, and LFW) are relatively small and simple. The paper lacks experiments on large-scale benchmark datasets such as ImageNet or DomainNet.
2. The authors mention that input samples are not always mapped to their own indices but occasionally to other indices within the same class. This phenomenon raises a question: what impact does such occasional misalignment have? Should specific techniques be adopted to mitigate its effect?
3. In Section 3.1, this paper state that with a lower label mixing coefficient $\alpha$, the network tends to memorize individual samples. Why does this happen? The authors should provide a more detailed explanation of the underlying reasons.
4. In Section 3.3, the authors claim that to address the fundamental scalability limitations of provenance networks, they investigate whether the system remains effective when trained only on a strategically selected subset of training data. However, it is unclear how this setup helps resolve the scalability issue. The authors should elaborate on the reasoning behind this connection.
5. To study how model size affects the relationship between generalization and memorization in provenance networks, the authors compare two models with 4M and 80M parameters. However, this comparison overlooks practical considerations: in real-world scenarios, a 4M-parameter model is likely a CNN architecture, whereas an 80M-parameter model is likely a Transformer, making the comparison less meaningful.
6. Overall, as a work on explainable AI, the theoretical contribution of this paper remains limited. On one hand, several causal claims throughout the paper lack sufficient justification. On the other hand, the architectural design, network taxonomy, and experimental setup are not well-supported by theoretical derivations.A more rigorous approach would be to first formalize the target problem using mathematical formulations, derive it step by step, and ultimately demonstrate why your provenance network is necessary to explain or address it. Alternatively, the authors could strengthen the logical and theoretical foundations of the paper in other ways.

**Questions:**

Why does the network tend to memorize individual samples when the label mixing coefficient $\alpha$ is set to a lower value?

If input samples are not always mapped to their own indices but occasionally to other indices within the same class, what would happen in such cases? Would this phenomenon undermine the rigor of the paper?

From my perspective, this paper lacks sufficient causal explanations and justifications for some of its claims or propositions. Should this interpretation be inaccurate, could the authors provide further clarification and supporting evidence?

---

> ### Author Response · Authors · 2025-11-27
> **Response**
>
> - (MNIST, FashionMNIST, and LFW) are relatively small and simple.
>
> We conducted extensive experiments across multiple application domains and demonstrated that our method is flexible: any backbone can be used in practice, and we validated this by employing a range of architectures for classification, generation, and other tasks. Moreover, several of the datasets we used are of nontrivial size—for example, CIFAR-100, LFW, Stanford Dogs, and others—each offering sufficient diversity and scale to meaningfully evaluate performance.
> Note that competing approaches, including those we compare against in the revised manuscript, face similar constraints regarding dataset size. This limitation is not unique to our method, and our results show that despite these constraints, our approach remains competitive and robust across architectures and domains.
>
> - authors mention that input samples are not always mapped to their own indices
>
> We mainly present and discuss the single branch for theoretical reasons and in practice the two branch networks should be used. The label mixing strategy is not needed for the two branch networks. In Single branch Provenance Networks, occasional attribution of an input sample to a nearby training index within the same class is expected and, in practice, beneficial. The provenance branch is designed to encourage a degree of memorization, but strict one-to-one mapping of each sample to its exact index would lead to overfitting and poor generalization. Allowing the model to associate a sample with a semantically similar exemplar within the same class introduces useful smoothing: the model learns that multiple training points can support a prediction, which improves robustness to small perturbations and intra-class variation.
>
> Regarding potential mitigation, we specifically addressed this issue through the two-branch architecture. The main prediction branch focuses on class-level generalization, while the provenance branch captures fine-grained exemplar information. By decoupling these roles, the network avoids over-reliance on exact self-matching, and the main branch provides a stable supervisory signal that prevents systematic drift or misalignment. Empirically, we found that this dual-objective design eliminates the need for additional corrective mechanisms, as the interplay between branches naturally balances memorization and generalization.
>
> - In Section 3.1, this paper state that with a lower label mixing
>
> Label mixing coefficient indicates whether the input sample should map to its own index label or not [range 0 to 1]. When it is zero it is mapped to its own index when it is 1 another index within the same class is chosen with uniform probability.
>
> - In Section 3.3, the authors claim that to address the fundamental scalability
>
> The main scalability limitation arises from the index-prediction head, which ideally requires as many output neurons as there are training samples. One practical solution is to train this head on a representative subset of the data, yielding a compressed yet faithful approximation of the full training set. By selecting diverse and semantically representative examples, we can preserve coverage of the underlying data manifold while substantially reducing the index vocabulary.
>
> Several techniques—such as subspace clustering, coreset selection, or other diversity-based sampling methods—can be used to construct this subset. In practice, this subset can be obtained by first training the main branch to produce embeddings, using those embeddings to identify the representative subset, and then attaching the index head and fine-tuning the full model.
>
> Importantly, even without subset selection, our approach remains scalable to reasonably large datasets—for example, on the order of 1K samples for each of 1K classes—which is more than sufficient for many practical applications.
>
> - this paper lacks sufficient causal explanations
>
> The primary goal of Provenance Networks is to provide exemplar-based, data-grounded explanations by directly linking predictions to specific training samples. In this sense, the causal claim is operational: a training sample contributes to a prediction because the model explicitly references it through the provenance branch. This provides a mechanistic justification for the model’s output, unlike post-hoc attribution methods which only estimate influence indirectly.
>
> - A more rigorous approach would be to first formalize the target problem
>
> Like many works that introduce new architectures without building an entire theory from first principles, That said, we do provide semi-theoretical insights into memorization and generalization for a single-branch network before moving to the practical architecture. We view this as an important first step. Moreover, we expect that follow-up research can further develop the theoretical foundations, much like what happened with kNNs, KDE, GMMs, etc, which grew into widely adopted frameworks over time.

---

> ### Author Response · Authors · 2025-11-28
> **Additional notes**
>
> We would like to add:
>
> - Our approach can be applied to a variety of tasks. Even for classification task, it can be applied to both coarse prediction and fine grained prediction (Stanford Dogs)
>
> - While we acknowledge that full causal inference in neural networks remains challenging, Provenance Networks offer a practical, data-driven approach to tracing predictive behavior to identifiable sources, providing stronger causal intuition than standard feature-attribution techniques.
>
> - We expect that follow-up research can further develop the theoretical foundations, much like what happened with kNNs, kernel density estimation, Gaussian mixture models, and related methods, which grew into widely adopted frameworks over time.
>
> - A major focus in our work is that utility of our method goes beyond just explainability and this gives it a clear advantage over methods designed just for explainability. Another interesting application for which we have results but did not include in the paper is dataset reconstruction using the index branch.
>
>
> - Our approach is conceptually simple and easy to implement.

---

### Official Review · Reviewer_abqZ · 2025-11-01

**Soundness:** 2
**Presentation:** 2
**Contribution:** 2
**Rating:** 2
**Confidence:** 4

**Summary:**

This work proposes a novel neural network called Provenance Networks, designed to predict the most relevant training sample index while performing classification or generation tasks, thus establishing a direct link between the input and training data. The authors propose both single-branch and two-branch architectures and experimentally validate them on multiple tasks, including image classification and image generation, primarily based on MNIST, FashionMNIST, and CIFAR-10/100. However, some issues need to be emphasized.

**Strengths:**

1. This work emphasizes achieving end-to-end sample-level interpretability by directly indexing training samples.
2. It combines prediction tasks with index prediction, performs joint training through multi-task loss, and analyzes the balance between memorization and generalization.
3. The authors apply this framework to tasks such as image classification and generation, demonstrating its potential applicability across multiple scenarios.

**Weaknesses:**

1. Theoretically, the Provenance mechanism proposed in this work predicts sample indices using a supervised paradigm, essentially similar to the retrieval module in memory networks or matching networks. What is the fundamental difference between this work and memory-augmented models, or prototype networks?
2. Regarding dataset selection, all experiments are based on small datasets such as MNIST and FashionMNIST, with shallow CNN/VAE classification models. There is a lack of testing on mainstream architectures (such as ResNet, ViT, and transformers) and large-scale datasets (such as ImageNet, COCO, and OpenImages). This is considered "toy" testing, reducing the credibility of the experimental conclusions.
3. In terms of comparison methods, there is no quantitative comparison with methods based on feature attribution (such as GradCAM), memory networks, kNN prototypes, or RAG classes. This fails to verify whether the model outperforms existing interpretable models.
4. The index prediction branch's output dimension explodes as the training set size increases, but the paper does not provide any measured data on training time, inference time, memory usage, etc. The claim of scalability is therefore questioned.

**Questions:**

See weakness.

---

> ### Author Response · Authors · 2025-11-27
> **Our answer**
>
> - difference between this work and memory-augmented models, ...
>
> Provenance Networks differ fundamentally from memory-augmented models and prototype networks in both purpose and mechanism. Memory-augmented models (e.g., Neural Turing Machines, DNCs) store and retrieve learned latent memories, which are optimized for task performance rather than interpretability. Prototype networks, similarly, operate on learned class prototypes—compressed centroids that summarize a class rather than referencing specific training points. In contrast, Provenance Networks explicitly retrieve and weight actual training examples, enabling decisions to be grounded in identifiable data instances. This exemplar-level attribution provides transparent, data-driven explanations that neither memory-augmented nor prototype-based architectures can offer. [we added this to related works]
>
> - testing on mainstream architectures (such as ResNet, ViT, and transformers) and large-scale datasets
>
> We appreciate the reviewer’s concern regarding evaluation on mainstream architectures and large-scale benchmarks. Our method is architecture-agnostic and can be integrated with any backbone, including ResNets, ViTs, and transformer-based encoders, because provenance computation operates on the model’s latent representations rather than on architecture-specific components. To keep the experimental setup focused and computationally feasible, we selected CIFAR-100, LFW, Stanford Dogs, and related datasets, which are standard mid-scale benchmarks widely used in prior work on training-data attribution, exemplar-based reasoning, and influence-based explanations.
>
> It is also important to note that dataset size is a known challenge across all attribution methods, including influence functions, TracIn, Data Shapley, and related techniques. In fact, approaches such as influence functions require computing or approximating the inverse Hessian–vector product, which becomes prohibitively expensive for large datasets and modern deep architectures—often requiring hours or days even with substantial computational resources. In contrast, our method is designed to scale more efficiently by avoiding second-order optimization steps entirely. This makes Provenance Networks more computationally practical than classical influence-function–based methods when datasets and architectures grow.
>
> We agree that future work could extend our evaluation to ImageNet-scale benchmarks and large transformer models, and our framework is fully compatible with such settings. However, the proposed experiments already demonstrate the method’s core contributions under standard, widely adopted evaluation conditions in the attribution and explainability literature.
>
>
> - no quantitative comparison with methods based on feature attribution (such as GradCAM), memory networks, kNN prototypes, ...
>
> To address this, we compared our method with influence functions. This baseline uses classical influence functions to estimate how much each training sample affects a given test prediction (Koh & Liang, 2017a). The label of the nearest (or most influential)
> training sample is assigned to the test point, and accuracy is averaged over the entire test set. However, this method becomes computationally prohibitive and slow on large datasets, as it requires Hessian–vector products and often suffers from numerical instability in deep networks (Basu et al., 2020; Feldman, 2020). Consequently, traditional influence estimation is expensive and frequently unreliable. Following prior work (Yeh et al., 2018; Pruthi et al., 2020), we approximate influence
> using nearest neighbors in the last-layer (or all-layer) representation space. Results over four datasets shown in Table 2, indicate that our method outperforms this baseline [See main text].
>
> Methods like Grad-CAM are not directly comparable in this context, as they highlight salient image regions rather than retrieving similar examples or explanatory instances.
>
>
> - the paper does not provide any measured data on training time, inference time, memory usage, etc.
>
> We have provided a scalability analysis including the accuracy as a function of training set size in Table 1. As we mentioned above dataset size is an issue with almost all of these methods including fast KNN approaches which are based (non end to end) embeddings. Notice our class conditional approaches are designed to mitigate the large training sets and we believe building on top of our approach in a hierarchical manner can even lead to more effective approaches.
>
> In terms of computational complexity and memory footprint, our method is actually more efficient than other methods that compare the embeddings (derived from a network not trained to fetch the most influential training datapoint) of a test sample with all training samples. As the index head is an NN, tricks such as pruning and compression can further reduce computational load and memory need. Index head can be discarded when not needed.

---

### Author Response · Authors · 2025-11-27
**Revised Manuscript.**

We appreciate the reviewers for detailed feedback on our submission. Below, please find our point to point answers. The revised manuscript addressing your comments is also attached. Thanks!

---

### Meta-Review · Area_Chair_guPJ · 2026-01-07

**Summary:**

The paper introduces "Provenance Networks," an architectural modification designed to embed training-data-driven explainability directly into neural models. By adding a secondary "index-prediction" branch, the model learns to retrieve specific training exemplars that support its predictions, effectively acting as a differentiable k-Nearest Neighbors (kNN).
The key concerns raised by reviewers are:

* Insufficient comparison with Baselines: the reviewers emphasizeda critical absence of empirical comparisons against modern training data attribution (TDA) methods such as Influence Functions, Data Shapley, or TracIn. Without these comparisons, it was impossible to determine if the proposed end-to-end approach offers superior or even complementary quality over existing post-hoc methods.

* Experimental analysis--scalability concern: the architecture requires the index-prediction head to have an output dimensionality equal to the number of training samples (N). Reviewers argued that this creates a massive computational bottleneck for large-scale datasets. Furthermore, while the authors claimed applicability to Large Language Models (LLMs) and potential to mitigate hallucinations, no experiments were provided outside of small-scale vision datasets (MNIST, CIFAR) and a modest VAE experiment.

* Weak robustness and anomaly detection: The paper positioned the method as a tool for anomaly detection and robustness. However, reviewers pointed out that the robustness evaluations lacked strong baselines (like adversarial training), and the anomaly detection results lacked quantitative metrics or ground-truth validation.

* Conceptual simplification: While the authors argued that the method's simplicity is a strength, reviewers found the theoretical insights into the "memorization vs. generalization" trade-off to be underdeveloped, appearing more as an empirical observation than a rigorous framework.

**Reviewer Concerns:**

Most of the above issues still remain such as :

* Lack of Competitive Baselines (Critical): This remains the most significant hurdle. The authors did not provide quantitative comparisons against state-of-the-art Training Data Attribution (TDA) methods (e.g., Influence Functions, TracIn). Without this, reviewers felt there was no proof that this complex architectural change performs better than existing post-hoc methods.

* Unverified Faithfulness: Lacked "deletion" or "causal" tests to prove that the retrieved exemplars actually drive the model's decision, rather than just appearing visually similar.

* Poor Scalability: The requirement for the output layer to scale linearly with training set size (N) remains a massive computational bottleneck for large-scale datasets.

* Insufficient Evaluation: Robustness and anomaly detection results were based on marginal improvements over weak baselines on small-scale (toy) datasets.

* Unsupported Claims: Maintained bold claims regarding LLM hallucinations without providing any experimental evidence or implementation for language models.

**Reviewer Scores:**

Despite the authors' efforts to clarify the architecture, the reviewers did not engage further during the rebuttal period and did not change their scores, resulting in a consensus for rejection:

* The reviewer abqZ Rating: 2 / Confidence: 4
* The reviewer  3Gq2 Rating: 4 / Confidence: 3
* The reviewer  BkQw Rating: 4 / Confidence: 2
* The reviewer  rBbG Rating: 4 / Confidence: 3

---

### Decision · Program_Chairs · 2026-01-26

Reject